# LSGQuant: Layer-Sensitivity Guided Quantization for One-Step Diffusion Real-World Video Super-Resolution

Tianxing Wu*[1]  Zheng Chen*[1]  Cirou Xu[1]  Bowen Chai[1]  Yong Guo[2]  Yutong Liu[†1]  Linghe Kong[1]
Yulun Zhang[†1]

## Abstract

One-step diffusion models have demonstrated a promising capability and fast inference in video super-resolution (VSR) for real-world applications. Nevertheless, the substantial model size and high computational cost of Diffusion Transformers (DiTs) limit downstream applications. While low-bit quantization is a common approach for model compression, the effectiveness of quantized models is challenged by the high dynamic range of input latent and diverse layer behaviors. To address these challenges, we introduce LSGQuant, a layer-sensitivity-guided quantization approach for one-step diffusion-based real-world VSR. Our method incorporates a Dynamic Range Adaptive Quantizer (DRAQ) to fit video token activations. Furthermore, we estimate layer sensitivity and implement a Variance-Oriented Layer Training Strategy (VOLTS) by analyzing layer-wise statistics in calibration. We also introduce Quantization-Aware Optimization (QAO) to jointly refine the quantized branch and a retained high-precision branch. Extensive experiments demonstrate that our method has nearly the performance of the original model with full-precision and significantly exceeds existing quantization techniques. All models and code are available at https://github.com/zhengchen1999/LSGQuant.

## 1. Introduction

As an essential and important computer vision task, video super-resolution (VSR) tries to reproduce high-quality videos with their original low-quality counterparts. With the fast promotion of smartphone video capture technologies

*Equal contribution [1]Shanghai Jiao Tong University [2]Huawei. Correspondence to: Yulun Zhang <yulun100@gmail.com>, Yutong Liu <isabelleliu@sjtu.edu.cn>.

*Proceedings of the 43rd International Conference on Machine Learning*, Seoul, South Korea. PMLR 306, 2026. Copyright 2026 by the author(s).

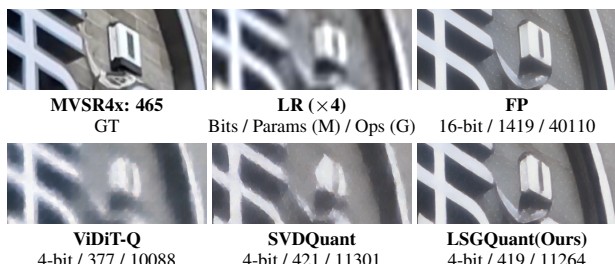

*Figure 1.* Visual comparison among the full-precision bfloat16 model, recent leading quantization method ViDiT-Q (Zhao et al., 2025), SVDQuant (Li et al., 2025a), and our low-bit LSGQuant.

and the widespread adoption of video streaming platforms, VSR has gained growing significance in practical applications. Initial VSR approaches (Jo et al., 2018; Nah et al., 2019; Liang et al., 2024) mainly concentrate on synthetic degradation scenarios, where low-resolution (LR) videos are produced using simple degradation assumptions. Nevertheless, such methods perform poorly in real-world environments with complex and unknown degradations.

To tackle these issues, a variety of approaches have been developed for real-world VSR. Among them, diffusion models can generate visually realistic details while maintaining stable training behavior (Ho et al., 2020; Blattmann et al., 2023; Zhou et al., 2024; Yang et al., 2025; Xie et al., 2025). Despite these advantages, DMs have high inference costs because of iterative denoising procedures. Although single-step diffusion models demonstrate strong ability in visual super-resolution (Chen et al., 2025; Wang et al., 2025), these models remain computationally heavy due to large-scale backbone architectures. Consequently, the inference cost of one-step diffusion VSR models is still relatively high, which restricts their real-world applicability.

Low-bit quantization serves as an effective approach to further lowering inference costs, due to its ability to significantly decrease computation costs. Through transferring floating-point values into low-precision, quantization facilitates model deployment on edge devices or platforms that have limited resources. However, applying low-bit quantization to diffusion-based video super-resolution models

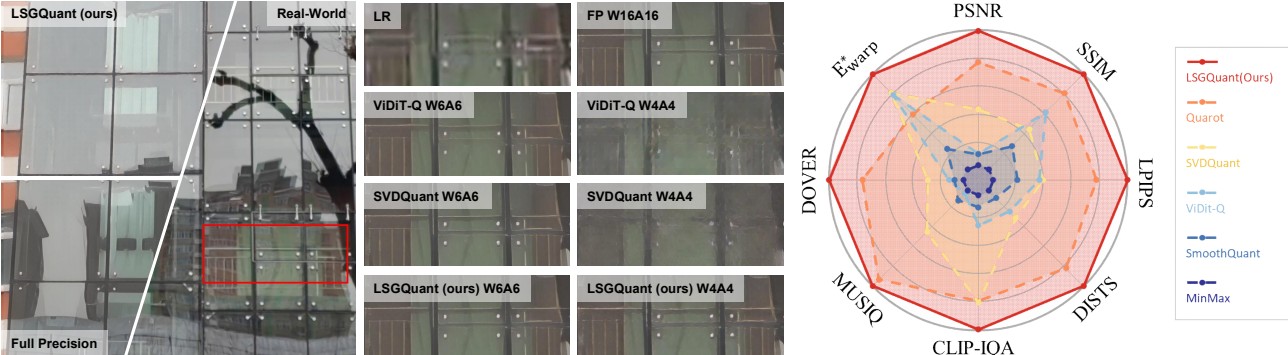

*Figure 2.* Performance comparisons on the synthetic MVSR4x dataset (Wang et al., 2023b) between different quantization methods in 4-bit and 6-bit scenarios. LSGQuant achieves leading scores on all evaluation metrics.

remains highly challenging. First, aggressive low-bit quantization often leads to severe performance degradation and noticeable loss in generation quality, particularly in video settings. Second, minimizing the quantization error during optimization often leads to overfitting the low-bit representation instead of optimal VSR performance.

Motivated by these challenges, we propose LSGQuant, a layer-sensitivity guided quantization framework for one-step diffusion real-world VSR. We select the prominent text-to-video (T2V) model WAN2.1 (Wan et al., 2025) as a full precision (FP) backbone, and adopt training settings of DOVE (Chen et al., 2025) for one-step diffusion VSR tasks. We apply several common methods in low-bit quantization, including Hadamard rotation (Ashkboos et al., 2024) and high-precision branch (Li et al., 2025a). Three novel components are introduced to address the specific challenges outlined earlier. Firstly, to handle the varying range, we introduce a Dynamic Range Adaptive Quantizer (DRAQ) to minimize quantization error. Second, our Variance-Oriented Layer Training Strategy (VOLTS) dynamically allocates training resources by re-estimating layer sensitivity, thereby enhancing output fidelity. Finally, we employ a Quantization-Aware Alternating Optimization (QAO) algorithm to refine quantized and full-precision branches.

Extensive experiments (*i.e.,* Figs 1 and 2) show that, under 4-bit quantization, LSGQuant has only negligible performance degradation and surpasses recent state-of-the-art quantization settings. When competing with the bfloat16 model, 4-bit LSGQuant achieves reductions of 70.49% in parameters (Params) and 71.92% in operations (Ops).

We list our contribution as follows:

- We introduce LSGQuant, an approach for one-step diffusion-based real-world VSR in model quantization. To the best of our knowledge, our approach firstly investigates a low-bit(*e.g.,* 4-bit) scenario.

- We design a Dynamic Range Adaptive Quantizer

(DRAQ) that effectively addresses the quantization error in the quantized inference process.

- In the calibration process, our Variance-Oriented Layer Training Strategy (VOLTS) and Quantization-Aware Alternating Optimization (QAO) provide reliable layer sensitivity estimation and quantization.

- Evaluations raised on both real-world and synthetic datasets demonstrate the advantageous behavior of our approach over existing quantization settings.

## 2. Related Work

### 2.1. Video Super-Resolution

Video super-resolution (VSR) originates from entering low-quality videos to get their high-resolution (HR) counterparts. Primitive VSR approaches (Jo et al., 2018; Chan et al., 2021; Liang et al., 2024) have achieved impressive performance. To better deal with unfixed degradation assumptions in the real world, suitable datasets and model architectures are mainly focused on. To construct more realistic datasets, RealBasicVSR (Chan et al., 2022) introduces diverse degradation models to synthesize training data, while MVSR4x (Wang et al., 2023b) collected LR-HR paired data from real environments. Besides, several real-world VSR methods enhance reconstruction quality by incorporating specialized architectural components. DiffVSR (Li et al., 2025b) imports multi-scale temporal attention for capturing temporal dependencies. Despite these significant advances, methods still have flaws in generating textures and details.

### 2.2. Diffusion Model

Diffusion models gain success in multiple computer tasks across both image (Ho et al., 2020; Ramesh et al., 2022; Rombach et al., 2022) and video (Blattmann et al., 2023; Chen et al., 2024; Zhou et al., 2024; Yang et al., 2025; Li

*Figure 3.* Overview of our LSGQuant. **Firstly**, we collect per-channel layer statistics by a single inference process. **Next**, we calculate layer sensitivity and estimate its importance to the final video output. **Finally**, we allocate training iterations by estimated sensitivities.

et al., 2025b; Xie et al., 2025; Wang et al., 2025) visual domains. To accelerate inference speed, one-step diffusion has been proposed as an extreme acceleration paradigm by decreasing inference steps from multi-step to 1. OSEDiff (Wu et al., 2024b) introduces one-step diffusion into image super-resolution via variational score distillation (Wang et al., 2023c), while DOVE (Chen et al., 2025) adopts high-quality fine-tuning with a latent-pixel strategy to adapt one-step diffusion models to video super-resolution. Despite significant speedup, one-step diffusion models remain computationally intensive due to large-scale architectures such as U-Net or Diffusion Transformer (DiTs).

### 2.3. Model Quantization

Quantization (Jacob et al., 2018) is a widespread lightning skill for neural network compression, aiming to reduce computational cost and memory footprint by representing floating-point values with low-bit numerical formats. To reduce the impact of outliers on quantization accuracy, equivalent transformations are widely used for reconciling quantization difficulty between activation and weights. SmoothQuant (Xiao et al., 2023) migrates activation outliers into weights through offline scaling, while QuaRot (Ashkboos et al., 2024) removes hidden-state outliers via Hadamard rotations. Recent studies have also explored quantization for diffusion models. SVDQuant (Li et al.,

2025a) proposes a high-precision but low rank and computation cost branch to compensate for quantization errors. The additional branch clearly fills the precision gap brought by quantization. Additionally, some quantization methods are designed for specific tasks, such as ViDiT-Q (Zhao et al., 2025) for visual generation and PassionSR (Zhu et al., 2025) for image super-resolution. Nevertheless, existing quantization models exhibit unsatisfactory performance when applied to video super-resolution based on one-step models.

### 3. Methodology

Our proposed LSGQuant will be fully introduced in this section, as shown in Fig. 3. First, we give preliminaries about quantization and previous methods based on scaling or rotation. Then, we describe our overall architecture for quantization layers and framework for the post-training quantization (PTQ) process. Finally, we give motivation and implementations for proposed methods: Dynamic Range Adaptive Activation Quantizer (DRAQ), Variance-Oriented Layer Training Strategy (VOLTS), and Quantization-Aware Alternating Optimization (QAO).

### 3.1. Preliminaries

**Model quantization** reduces computational and memory footprints by using low-bit integers to replace floating-point,

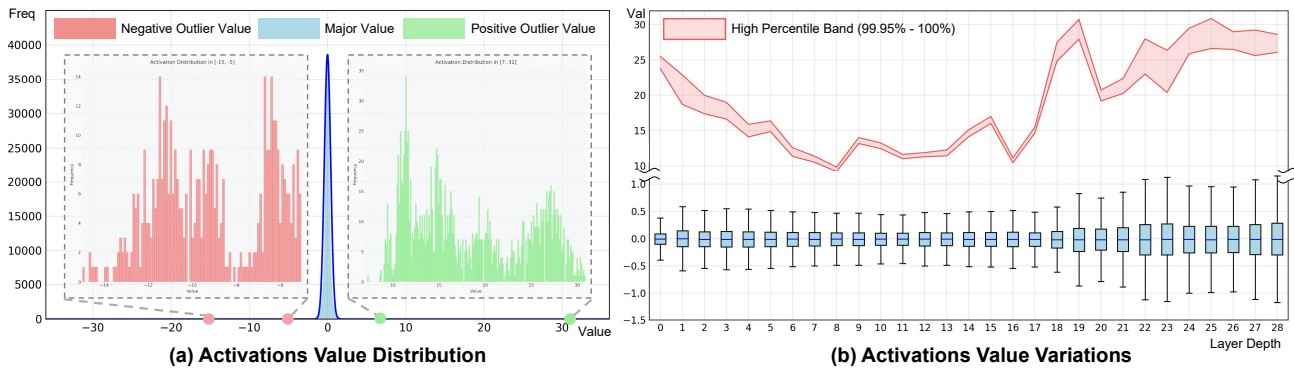

**(a) Activations Value Distribution**    **(b) Activations Value Variations**

*Figure 4.* Input data sampled from the cross-attention query layer. (a) Statistical distribution of activation values. While the majority of values are concentrated near zero, a significant long-tail distribution exists. These extreme values would dominate the quantization range. (b) Distribution of input latent values across different blocks. The statistical characteristics of outliers vary with depth.

accompanied by high-bit parameters for efficiency. Given an input floating-point value $x$, the quantization and inverse procedures can be formally expressed as:

$$x_{int} = \text{Clip}(\lfloor \frac{x}{s} \rceil - z, l, u), \hat{x} = s \cdot (x_{int} + z), \quad (1)$$

where $x_{int}$ is the quantized integer value bounded by the clip boundary $l$ and $u$, which are determined by the quantization bit-width and symmetric type. $s$ and $z$ denote scaling factors and zero points, respectively.

**Pre-scaling-based methods** (Xiao et al., 2023; Wu et al., 2024a; Zhao et al., 2025) uses a channel-wise scaling factor $\mathbf{s} \in \mathbb{R}^c$ to migrates activation outliers into weights:

$$s_i = \frac{\max(|\mathbf{X}_{:,i}|)^\alpha}{\max(|\mathbf{W}_{:,i}|)^{1-\alpha}}, i \in \{1, 2, \cdots, C\},$$
$$\mathbf{s} = \text{diag}(s_1, s_2, \cdots, s_C), \quad (2)$$
$$\mathbf{Y} = Q_A(\mathbf{X}\mathbf{s}^{-1}) \cdot Q_W(\mathbf{s}\mathbf{W}) + \mathbf{b},$$

where $\mathbf{C}$ is used as the input channel number, and $\alpha$ is a hyperparameter used to control the degree of outlier transfer. This parameter can be finetuned on the calibration dataset, or set to a uniform value across all layers.

**Rotation-based methods** (Ashkboos et al., 2024; Liu et al., 2025; Zhao et al., 2025) employs rotation matrix $\mathbf{Q}$ which satisfies $\mathbf{Q}\mathbf{Q}^T = \mathbf{I}$ and $|\mathbf{Q}| = \mathbf{I}$. $\mathbf{Q}$ can be generated by fast-hadamard-transform or trained on manifold. By respectively applying $\mathbf{Q}$ and $\mathbf{Q}^\mathbf{T}$ on weights and activations, large outliers are effectively suppressed.

### 3.2. Overall Framework

LSGQuant adopts WAN2.1 (Wan et al., 2025), a multi-step diffusion model, as the backbone network. Following the settings same as DOVE (Chen et al., 2025), we train a one-step version for VSR tasks. The diffusion transformer (DiT) module dominates the computational costs in the inference process; therefore, we focus on quantizing all linear modules in DiT by replacing the original layers with quantized layers.

The whole architecture of the quantization layer is shown in Fig 3(c). The layer has two main branches, including a low-bit quantized branch and a full-precision (FP) low-rank branch. The entire formulation for the layer is:

$$\mathbf{Y} = \mathbf{X}\mathbf{W} + \mathbf{b} \approx \underbrace{\mathbf{X}\mathbf{L_1}\mathbf{L_2}}_{\text{FP, low-rank}} + \underbrace{Q_A(\mathbf{X}\mathbf{H})\,Q_W(\mathbf{H}\mathbf{W_R})}_{\text{low-bit quantized}} + \mathbf{b}.$$
$$(3)$$

In the full-precision and low-rank branch, we apply two matrices $\mathbf{L_1} \in \mathbb{R}^{m \times r}$ and $\mathbf{L_2} \in \mathbb{R}^{r \times n}$ for limiting computational costs. In the low-bit branch, $Q_W$ and $Q_A$ refer to different quantizers for weights and activation values, and $\mathbf{W_R} = \mathbf{W} - \mathbf{L_1}\mathbf{L_2}$ to assure residual acts relatively complementary with the other branch. Following previous work, $\mathbf{H}$ is the random Hadamard matrix, and applying these matrices can be computed through fast-hadamard-transform.

The calibration process comprises two important stages. Firstly, we analyze the layer sensitivity on the calibration dataset. We run a full inference process, collect layer statistics, and allocate different training importance. Secondly, for each linear layer, we run the adaptive optimizing algorithm with different epochs decided in the last step. After the optimization, we confirm required parameters for $\mathbf{L_1}, \mathbf{L_2}$ and $\mathbf{W_R}$ used for quantization inference process.

### 3.3. Dynamic-Range Adaptive Activation Quantizer

Activation distributions in Diffusion Transformers (DiTs) exhibit significant channel-wise imbalance. As illustrated in Fig. 4(a), a small number of channels contain extreme outlier values that dominate the overall quantization range. Consequently, conventional min-max quantization suffers from severe resolution loss, especially under low-bit (e.g., 4-bit) constraints. Pre-scaling methods also face challenges because calibration datasets may not capture the full dynamic range of activations encountered during inference, leading to persistent quantization errors. Furthermore, quantization strategies designed for multi-step denoising can optimize mi-

gration degree $\alpha$ by aggregating statistics across timesteps. However, this approach is ineffective for one-step inference, where activations are generated from a single timestep and cannot be smoothed through temporal averaging.

To overcome these challenges, we propose the **Dynamic-Range Adaptive Activation Quantizer (DRAQ)**. For an activation tensor $X_{N \times C}$, which represents token numbers $\times$ channel dimension, we first compute the channel-wise maximum absolute value $S$ across all tokens, which provides a robust estimate of the per-channel dynamic range. Activations are then normalized by these estimated scales to the interval $[-1, 1]$. This normalization is friendly for symmetric quantization and suppresses outliers from a token-wise perspective. Finally, the quantized values are rescaled to the original range, ensuring compatibility with subsequent layers and subsequent calculations.

As shown in Fig. 3(a), the whole quantization process can be formulated as follows:

$$s_i = \max_N |\mathbf{X}_{N,i}|, \mathbf{s} = \text{diag}(s_1, \ldots, s_C),$$

$$\tilde{\mathbf{X}} = \mathbf{X}\mathbf{s}^{-1}, d_i = \max_C |\tilde{\mathbf{X}}_{i,C}|, \mathbf{d} = \text{diag}(d_1, \ldots, d_N),$$

$$\hat{\mathbf{X}} = \mathbf{d} \, \text{Clamp}\left(\left\lfloor (2^{n-1}-1)\mathbf{d}^{-1}\tilde{\mathbf{X}} \right\rceil, -2^{n-1}, 2^{n-1}-1\right)\mathbf{s}, \tag{4}$$

where $n$ denotes the quantization bitwidth, $\tilde{\mathbf{X}}$ and $\hat{\mathbf{X}}$ denotes the scaled input $\mathbf{X}$ and quantized $\mathbf{X}$ respectively.

The scaling factors are computed online with negligible extra computational and memory costs. By normalizing activations with respect to their channel-wise maximum magnitudes, DRAQ directly adapts the quantization range to the activation statistics and aligns the effective dynamic ranges, thereby reducing quantization errors.

### 3.4. Variance-Oriented Layer Training Strategy

Existing quantization methods for Diffusion Transformers (DiTs) are designed for multi-step denoising (Shang et al., 2023; He et al., 2024) and assume that all network layers contribute equally to output quality. This assumption becomes invalid for one-step diffusion models, where the entire latent output is generated in a single forward pass with a fixed timestep, and there is no opportunity for iterative steps to correct quantization errors.

Furthermore, one-step diffusion models are typically initialized and finetuned from large pretrained diffusion models (Wang et al., 2024; Dong et al., 2025; Chen et al., 2025), and different network components exhibit highly heterogeneous response behaviors with respect to varying input latent. As illustrated in Figure 4(b), we observed a clear disparity in activation variability across layers. As a result, applying a uniformly quantization method on all network components is a suboptimal method for output quality.

To address this issue, we propose to allocate optimization effort according to each layer's sensitivity to input variations. Specifically, we introduce **Variance-Oriented Layer Trining Strategy (VOLTS)**, which reallocates layer-wise training budgets based on activation statistics, without modifying the model architecture or loss function.

VOLTS consists of a calibration phase followed by variance-guided training budget allocation as shown in Fig 3(b):

**Step 1: Layer Statistics Extraction.** Inspired by SmoothQuant (Xiao et al., 2023) and ViDiT-Q (Zhao et al., 2025), for each linear layer $l$, we collect its input activation $\mathbf{X}^l \in \mathbb{R}^{B \times N \times C}$, then we compute its channel-wise mean as a compact representation of the layer-level activation drift:

$$\mu^l(\mathbf{X}) = \frac{1}{C} \sum_{i=1}^{C} \mathbf{X}^l_{:,:,i}, \tag{5}$$

Compared to extreme-value statistics, the channel mean better reflects semantic-level representation changes while being robust to outliers, which is a better estimation metric.

**Step 2: Layer sensitivity estimation.** Using a calibration dataset $\mathcal{D}_{\text{calib}}$, we estimate the variance

$$\sigma_l^2 = \text{Var}_{\mathbf{X} \sim \mathcal{D}_{\text{calib}}}(\mu^l(\mathbf{X})), \tag{6}$$

for each layer. This variance measures how strongly a layer's representation responds to different inputs. Layers with higher variance are more sensitive to data distribution changes and are therefore more vulnerable to quantization-induced errors, which need more training resources.

**Step 3: Variance-guided training budget allocation.** Based on two thresholds $\delta_1$ and $\delta_2$, layers are categorized into three groups for different training strategies:

- $\sigma_l^2 \in [0, \delta_1)$: Low-variance layers, whose representations remain nearly invariant across inputs, are **frozen** after quantization layer initialization during training to avoid unnecessary disturbance.

- $\sigma_l^2 \in [\delta_1, \delta_2)$: Medium-variance layers are assigned limited training process for **light adaptation**.

- $\sigma_l^2 \in [\delta_2, +\infty)$: High-variance layers, exhibiting substantial input-dependent changes, are **optimized** until convergence to compensate for quantization errors.

By reallocating training budgets toward variance-sensitive layers, VOLTS prioritizes perceptually critical components of the network. In one-step diffusion models, variance-guided layer-wise adaptation shows particular effectiveness because minimizing global quantization error does not necessarily correlate with perceptual quality.

### 3.5. Quantization-Aware Alternating Optimization

In low-bit quantization scenarios, the limited numerical precision is used to represent floating-point weights and acti-

vations, which inevitably leads to performance degradation. To mitigate the gap, a widely adopted strategy is to introduce a high-precision branch alongside the quantized branch (He et al., 2024; Li et al., 2025a; Fu et al., 2025). However, these two branches are often strongly coupled and have nearly identical influence on the final output. Merely enhancing the high-precision branch can inadvertently degrade the performance of the low-bit branch, and vice versa. This strong coupling necessitates a more principled joint optimization strategy that accounts for quantization constraints.

To address these issues, we propose **Quantization-aware Alternating Optimization (QAO)**. We formulate the quantization objective as follows:

$$\min_{\mathbf{L_1},\mathbf{L_2}} \|(\hat{\mathbf{W}} - \mathbf{L_1}\mathbf{L_2}) - \mathrm{Q_W}(\hat{\mathbf{W}} - \mathbf{L_1}\mathbf{L_2})\|_F \quad (7)$$

Here, the low-rank branch is optimized to approximate the quantization-compensated residual, rather than the original weight tensor directly. When training from scratch, the overall loss converges slowly. Moreover, if we apply the straight-through estimator (STE) gradient

$$\frac{\partial\, \mathrm{Q}(x)}{\partial x} = \mathbf{1}[x \in [l, u]], \quad (8)$$

the gradient can vanish when $\hat{\mathbf{W}} - \mathbf{L_1}\mathbf{L_2}$ becomes sufficiently small, halting further optimization. Inspired by SVDQuant, we first perform singular value decomposition (SVD) on $\hat{\mathbf{W}}$ to obtain a strong initialization for the low-rank parameters and fix quantization parameters, thereby avoiding repeated searches for optimal quantization scales. While this provides a favorable starting point, both SVD and quantization introduce inevitable approximation errors. Focusing solely on minimizing one type of error tends to yield suboptimal solutions in the quantized latent space. As outlined in Algorithm 1, we reduce quantization error by iteratively computing the residual $\hat{\mathbf{W}} - \mathrm{Q_W}(\hat{\mathbf{W}} - \mathbf{L_1}\mathbf{L_2})$ and adjusting $\mathbf{L_1}$ and $\mathbf{L_2}$ over several iterations, finally selecting the parameters that achieve the minimal loss.

## 4. Experiments

### 4.1. Experimental Settings

**Data construction.** We use the HQ-VSR (Chen et al., 2025) dataset for both one-step model training and calibration process. In the calibration stage, we randomly sample 50 videos and run the full inference process using the FP DiT model. Evaluation is conducted on synthetic and real-world datasets. UDM10 (Tao et al., 2017) and REDS30 (Nah et al., 2019) encompass various degradation types, which are well-suited for synthetic datasets. We employ the MVSR4x (Wang et al., 2023b) dataset for real-world evaluations.

**Evaluation Metrics.** We assess model performance using quality assessment metrics for image (IQA) and video (VQA). The reference-based IQA metrics we adopt contain DISTS (Ding et al., 2020), PSNR, SSIM (Wang et al.,

---

**Algorithm 1** Quantization-Aware Alternating Optimization

1: **Input:** Layer weight $\mathbf{W}$, rank $r$, iteration rounds $n$
2: $\hat{\mathbf{W}} \leftarrow \mathbf{W}\mathbf{H}$
3: $[\mathbf{U}, \boldsymbol{\Sigma}, \mathbf{V}] \leftarrow \mathrm{SVD}(\hat{\mathbf{W}})$
4: $\mathbf{L_1} \leftarrow \mathbf{U}\boldsymbol{\Sigma}_{:,:r}, \mathbf{L_2} \leftarrow \mathbf{V}_{:r,:}$
5: $\mathbf{R} \leftarrow \hat{\mathbf{W}} - \mathbf{L_1}\mathbf{L_2}, \mathbf{W_R} \leftarrow \mathrm{Q_W}(\mathbf{R})$
6: $Err^* \leftarrow +\infty$
7: $\mathbf{L_1^*} \leftarrow \emptyset, \mathbf{L_2^*} \leftarrow \emptyset$
8: **for** $i = 1$ **to** $n$ **do**
9:    $err \leftarrow \|\mathbf{R} - \mathbf{W_R}\|_F$
10:   **if** $err < Err^*$ **then**
11:      $Err^* \leftarrow err$
12:      $\mathbf{L_1^*} \leftarrow \mathbf{L_1}, \mathbf{L_2^*} \leftarrow \mathbf{L_2}$
13:   **end if**
14:   $[\mathbf{U}, \boldsymbol{\Sigma}, \mathbf{V}] \leftarrow \mathrm{SVD}(\hat{\mathbf{W}} - \mathbf{W_R})$
15:   $\mathbf{L_1} \leftarrow \mathbf{U}\boldsymbol{\Sigma}_{:,:r}, \mathbf{L_2} \leftarrow \mathbf{V}_{:r,:}$
16:   $\mathbf{R} \leftarrow \hat{\mathbf{W}} - \mathbf{L_1}\mathbf{L_2}, \mathbf{W_R} \leftarrow \mathrm{Q_W}(\mathbf{R})$
17: **end for**
18: **return** $\mathbf{W_R}, \mathbf{L_1^*}, \mathbf{L_2^*}$

---

2004) and LPIPS (Zhang et al., 2018). We also utilize MANIQA (Yang et al., 2022), CLIP-IQA (Wang et al., 2023a) and MUSIQ (Ke et al., 2021), considering their no-reference based application. For video quality assessment, we employ warping error $E^*_{warp}$ (Lai et al., 2018) based on flows and DOVER (Wu et al., 2023).

**Implementation Details.** Following the training settings of DOVE (Chen et al., 2025), we trained an one-step inference version of WAN2.1 (Wan et al., 2025) as basic full-precision(FP) backbone. The hyperparameters $\delta_1$ and $\delta_2$ for VOLTS are set to 0.001 and 0.075. The number of iterations for frozen, light adaptation, and fully optimized are respectively set to 1, 30, and $+\infty$. For weight quantization, we use a static asymmetric channel-wise quantizer. The rank for the SVD path is set to $r = 32$. We use PyTorch and an NVIDIA RTX A6000 GPU to build all experiments.

### 4.2. Comparison with State-of-the-Art Methods

We employ several quantization methods for approach comparison: MinMax (Jacob et al., 2018), SmoothQuant (Xiao et al., 2023), Quarot (Ashkboos et al., 2024), ViDiT-Q (Zhao et al., 2025), and SVDQuant (Li et al., 2025a). We perform the qualitative and quantitative results with comparison aspects. We also compute the compression ability of our quantized model compared with the original FP backbone to fully demonstrate our performance in low-bit quantization.

**Qualitative Results.** Tab 1 exhibits quantitative values and comparisons. Our method transcends prior approaches on almost every metric in 4-bit settings. Under 6-bit conditions, the quantization error is reduced, narrowing the gap in reconstruction quality among various methods. No-reference

*Table 1.* Quantitative results on synthetic and real-world datasets. WAN stands for our one-step inference version for VSR tasks based on training settings of DOVE(Chen et al., 2025), with pretrained WAN2.1 (Wan et al., 2025) as our FP backbone components. Red and Blue texts represent the best and second-best scores, respectively.

| Dataset | Bit | Method | PSNR↑ | SSIM↑ | LPIPS↓ | DISTS↓ | CLIP-IQA↑ | MUSIQ↑ | MANIQA↑ | DOVER↑ | $E^*_{warp}$↓ |
|---|---|---|---|---|---|---|---|---|---|---|---|
| UDM10 | W16A16 | WAN (Wan et al., 2025) | 23.74 | 0.7144 | 0.2886 | 0.2080 | 0.4973 | 63.85 | 0.3482 | 0.5494 | 0.93 |
| | W6A6 | MinMax (Jacob et al., 2018) | 23.27 | 0.7090 | 0.2987 | 0.2154 | 0.5205 | 63.73 | 0.3412 | 0.5135 | 0.95 |
| | | SmoothQuant (Xiao et al., 2023) | 23.02 | 0.7014 | 0.3011 | 0.2180 | 0.5237 | 64.39 | 0.3520 | 0.5579 | 0.89 |
| | | QuaRot (Ashkboos et al., 2024) | 23.66 | 0.7128 | 0.2912 | 0.2120 | 0.4990 | 63.78 | 0.3491 | 0.5403 | 0.92 |
| | | ViDiT-Q (Zhao et al., 2025) | 23.08 | 0.7024 | 0.3005 | 0.2167 | 0.5224 | 64.38 | 0.3499 | 0.5513 | 0.86 |
| | | SVDQuant (Li et al., 2025a) | 23.71 | 0.7125 | 0.2906 | 0.2101 | 0.4931 | 63.50 | 0.3441 | 0.5342 | 0.95 |
| | | LSGQuant (ours) | 23.69 | 0.7126 | 0.2911 | 0.2096 | 0.5028 | 64.00 | 0.3486 | 0.5441 | 0.94 |
| | W4A4 | MinMax (Jacob et al., 2018) | 19.68 | 0.6200 | 0.5789 | 0.4482 | 0.2568 | 28.97 | 0.2078 | 0.0448 | 2.12 |
| | | SmoothQuant (Xiao et al., 2023) | 20.60 | 0.6288 | 0.5372 | 0.4233 | 0.2596 | 32.29 | 0.2189 | 0.0446 | 2.24 |
| | | QuaRot (Ashkboos et al., 2024) | 22.72 | 0.6941 | 0.3489 | 0.2672 | 0.4346 | 56.90 | 0.2775 | 0.3436 | 1.36 |
| | | ViDiT-Q (Zhao et al., 2025) | 21.89 | 0.6570 | 0.5052 | 0.4012 | 0.2565 | 25.72 | 0.1962 | 0.0559 | 1.36 |
| | | SVDQuant (Li et al., 2025a) | 21.92 | 0.6544 | 0.4770 | 0.3781 | 0.5007 | 46.48 | 0.2768 | 0.1718 | 1.23 |
| | | LSGQuant (ours) | 23.80 | 0.7130 | 0.3169 | 0.2348 | 0.4966 | 59.15 | 0.3139 | 0.4333 | 1.07 |
| REDS30 | W16A16 | WAN (Wan et al., 2025) | 19.42 | 0.5145 | 0.3182 | 0.2479 | 0.3256 | 61.86 | 0.2889 | 0.4073 | 2.23 |
| | W6A6 | MinMax (Jacob et al., 2018) | 19.14 | 0.5129 | 0.3412 | 0.2653 | 0.3082 | 60.22 | 0.2784 | 0.3715 | 2.17 |
| | | SmoothQuant (Xiao et al., 2023) | 19.03 | 0.5087 | 0.3379 | 0.2645 | 0.3179 | 61.71 | 0.2866 | 0.3975 | 2.07 |
| | | QuaRot (Ashkboos et al., 2024) | 19.36 | 0.5152 | 0.3269 | 0.2551 | 0.3226 | 61.29 | 0.2866 | 0.3910 | 2.11 |
| | | ViDiT-Q (Zhao et al., 2025) | 19.14 | 0.5100 | 0.3399 | 0.2667 | 0.3155 | 61.26 | 0.2831 | 0.3897 | 1.96 |
| | | SVDQuant (Li et al., 2025a) | 19.39 | 0.5134 | 0.3220 | 0.2494 | 0.3271 | 61.23 | 0.2862 | 0.3930 | 2.29 |
| | | LSGQuant (ours) | 19.37 | 0.5128 | 0.3197 | 0.2491 | 0.3332 | 61.82 | 0.2884 | 0.4006 | 2.31 |
| | W4A4 | MinMax (Jacob et al., 2018) | 18.16 | 0.4595 | 0.6475 | 0.4835 | 0.2442 | 28.55 | 0.2204 | 0.0500 | 4.62 |
| | | SmoothQuant (Xiao et al., 2023) | 18.74 | 0.4706 | 0.6161 | 0.4844 | 0.2726 | 28.81 | 0.2113 | 0.0468 | 4.44 |
| | | QuaRot (Ashkboos et al., 2024) | 18.65 | 0.4921 | 0.4354 | 0.3445 | 0.2990 | 50.88 | 0.2368 | 0.2081 | 3.23 |
| | | ViDiT-Q (Zhao et al., 2025) | 19.15 | 0.4830 | 0.5990 | 0.4607 | 0.2363 | 24.57 | 0.2085 | 0.0463 | 3.81 |
| | | SVDQuant (Li et al., 2025a) | 18.70 | 0.4841 | 0.5387 | 0.4446 | 0.3535 | 32.61 | 0.2390 | 0.1022 | 2.70 |
| | | LSGQuant (ours) | 19.72 | 0.5138 | 0.3671 | 0.2822 | 0.3067 | 54.31 | 0.2521 | 0.2967 | 2.73 |
| MVSR4x | W16A16 | WAN (Wan et al., 2025) | 22.80 | 0.7434 | 0.3353 | 0.2661 | 0.4794 | 63.49 | 0.3904 | 0.5020 | 0.46 |
| | W6A6 | MinMax (Jacob et al., 2018) | 22.68 | 0.7460 | 0.3373 | 0.2712 | 0.4878 | 62.79 | 0.3771 | 0.4606 | 0.43 |
| | | SmoothQuant (Xiao et al., 2023) | 22.56 | 0.7438 | 0.3374 | 0.2724 | 0.4870 | 63.13 | 0.3885 | 0.4915 | 0.41 |
| | | QuaRot (Ashkboos et al., 2024) | 22.75 | 0.7438 | 0.3401 | 0.2693 | 0.4759 | 63.47 | 0.3907 | 0.4911 | 0.44 |
| | | ViDiT-Q (Zhao et al., 2025) | 22.56 | 0.7435 | 0.3365 | 0.2740 | 0.4811 | 62.92 | 0.3851 | 0.4939 | 0.39 |
| | | SVDQuant (Li et al., 2025a) | 22.79 | 0.7419 | 0.3374 | 0.2689 | 0.4778 | 63.62 | 0.3893 | 0.4957 | 0.47 |
| | | LSGQuant (ours) | 22.79 | 0.7420 | 0.3337 | 0.2655 | 0.4790 | 63.74 | 0.3903 | 0.5072 | 0.47 |
| | W4A4 | MinMax (Jacob et al., 2018) | 20.89 | 0.7139 | 0.5288 | 0.4203 | 0.2549 | 25.35 | 0.3416 | 0.0661 | 1.07 |
| | | SmoothQuant (Xiao et al., 2023) | 21.05 | 0.7226 | 0.4934 | 0.4091 | 0.2728 | 28.73 | 0.2261 | 0.0891 | 0.95 |
| | | QuaRot (Ashkboos et al., 2024) | 22.37 | 0.7422 | 0.3773 | 0.2990 | 0.4121 | 56.70 | 0.2988 | 0.3357 | 0.75 |
| | | ViDiT-Q (Zhao et al., 2025) | 21.06 | 0.7351 | 0.4613 | 0.3877 | 0.2997 | 26.34 | 0.2307 | 0.1046 | 0.63 |
| | | SVDQuant (Li et al., 2025a) | 21.69 | 0.7289 | 0.4559 | 0.3780 | 0.4141 | 39.80 | 0.2717 | 0.1615 | 0.62 |
| | | LSGQuant (ours) | 22.82 | 0.7492 | 0.3317 | 0.2715 | 0.4543 | 58.90 | 0.3529 | 0.4257 | 0.52 |

*Table 2.* A comparison of Params, Ops, and compression ratios (DiT module only) across various quantization settings. Ops are computed using a latent input size of 16×9×90×158, which represents a 32x3×180×318 video in the UDM10 dataset.

| Method | Bits | Params / M (↓ Ratio) | Ops / G (↓ Ratio) |
|---|---|---|---|
| WAN2.1 | W16A16 | 1419 (↓0%) | 40110(↓0%) |
| LSGQuant | W8A8 | 766 (↓45.97%) | 21284 (↓46.94%) |
| | W6A6 | 592 (↓58.23%) | 16274 (↓59.43%) |
| | W4A4 | 419 (↓70.49%) | 11264 (↓71.92%) |

IQA and VQA metrics are more sensitive to subtle texture differences and stylistic shifts, thereby amplifying fluctuations in perceptual scores. Nevertheless, the consistent improvement of reference-based IQA metrics indicates that our method still has advantages in restoring real structural information and suppressing low-bit quantization distortion.

**Visual Results.** Figs. 5 and 6 present visual comparisons. Compared to prior quantization approaches, LSGQuant produces more faithful textures and clearer details, with min-imal difference from the FP model. When applied to Diffusion Transformer, previous leading quantization methods could generate impractical artifacts (e.g., SVDQuant on REDS30: 003), while our LSGQuant alleviates these distortions. Moreover, due to the special design of the activation quantizer, LSGQuant has higher temporal consistency and better dynamic continuity. This is attributed to the combined effects of different mechanisms designed in both the inference process and training strategy.

**Compression Ability.** Following previous work (Qin et al., 2023), we report parameters (Params/M) and computational complexity (Ops/G) to demonstrate compression ratio. As shown in Table 2, versus the FP backbone, 6-bit LSGQuant reduces module params and computing by 58.08% and 59.43%, while 4-bit cuts them by 70.34% and 71.92%. These results fully demonstrate the compression capability and deployment potential of our method.

**Thresholds sensitivity.** We further examine the sensitivity-

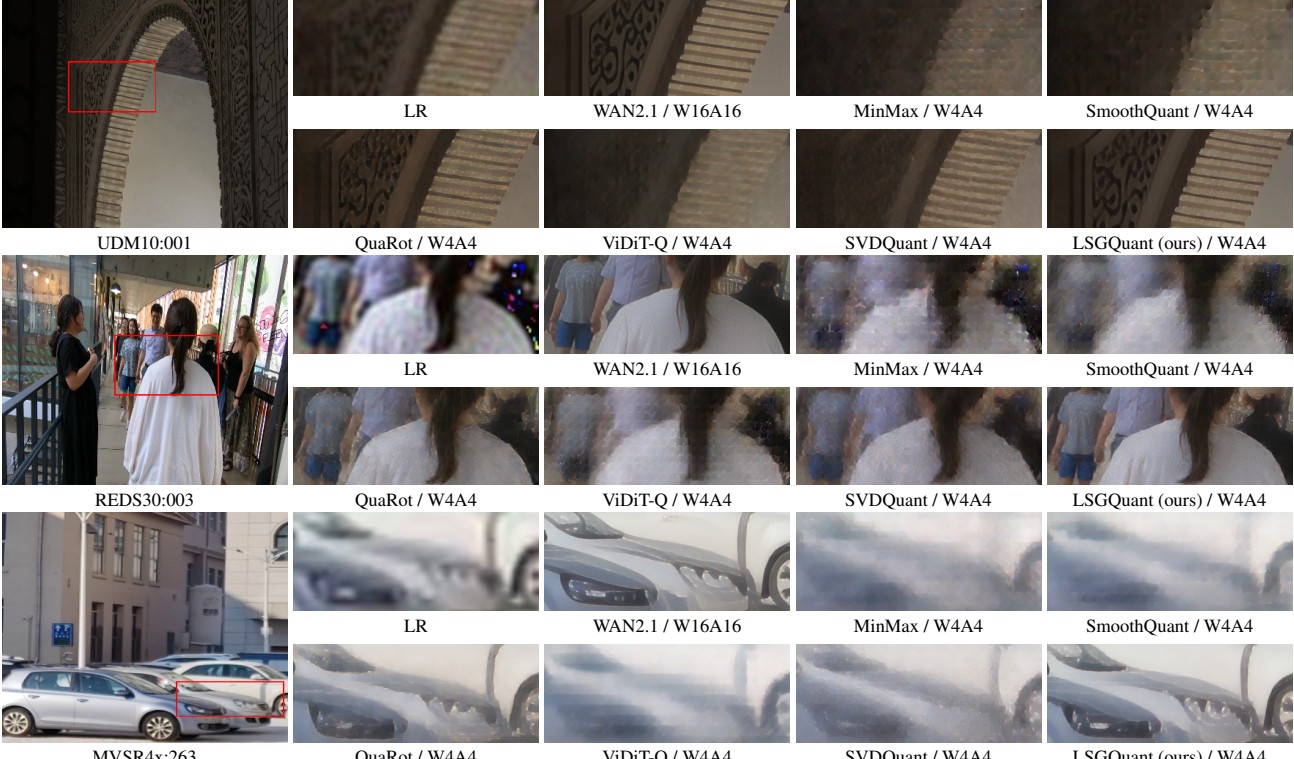

*Figure 5.* Visual comparison on synthetic UDM10 (Tao et al., 2017), REDS30 (Nah et al., 2019) and real-world MVSR4x (Wang et al., 2023b) datasets at 4-bit quantization. Our approach outperforms existing methods in the 4-bit setting.

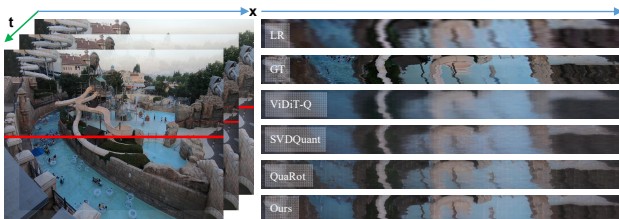

*Figure 6.* Comparison of temporal consistency (by stacking the red line across frames). Our method achieves better spatial and temporal consistency than the compared approaches.

based clustering of the 300 linear modules in the DiT blocks. The per-layer sensitivity scores $\sigma_l^2$ naturally form three well-separated clusters with clear gaps:

- A low-sensitivity cluster comprising 90 layers ($\sigma_l^2 \in [0, 3.03 \times 10^{-5}]$), which includes all the k, v, and o projections of the cross-attention modules;

- A medium-sensitivity cluster of 193 layers ($\sigma_l^2 \in [1.88 \times 10^{-3}, 0.069]$), covering most of the remaining modules in the DiT blocks;

- A high-sensitivity cluster with 17 layers ($\sigma_l^2 \in [0.089, 0.369]$), predominantly consisting of self-attention modules located near the beginning and end of the DiT blocks.

Because of the substantial gaps between these clusters, the thresholds $\delta_1$ and $\delta_2$ can be chosen arbitrarily within the gaps without changing the layer classification, making the three-tier scheme insensitive to the exact threshold values. This clustering directly motivates the three distinct training strategies adopted in VOLTS.

### 4.3. Ablation Study

**Dynamic-Range Adaptive Activation Quantizer (DRAQ).** We conduct experiments under three activation quantization settings: (1) a baseline per-token symmetric quantizer without channel-wise range adaptation; (2) a static channel-wise quantizer that uses fixed ranges, determined by collecting per-channel absolute maximum values from the calibration dataset concurrently with Step 1 of VOLTS; and (3) our proposed DRAQ. As shown in Tab. 3a, DRAQ yields significant performance gains compared to the other two settings. Specifically, the performance drops significantly if the scaling factor is replaced with data determined in setting (2), which further demonstrates that calibration data fails to capture the complete input dynamic range. So applying a flexible and dynamic activation quantization strategy is significant for minimizing quantization error. These experiments evaluate the effectiveness of DRAQ.

**Variance-Oriented Training Strategy (VOLTS).** We al-

*Table 3.* Ablation study on different method settings about DRAQ, VOLTS, and QAO. Experiments are conducted on the dataset HQ-VSR (Chen et al., 2025) with full-precision inference, and evaluated on the synthetic dataset UDM10 (Tao et al., 2017).

*(a)* Activation quantizer scaling calculation.

| Method | No scaling | Calibrated Scaling | DRAQ |
|---|---|---|---|
| PSNR ↑ | 23.72 | 20.04 | **23.80** |
| DISTS ↓ | 0.2380 | 0.4352 | **0.2348** |
| CLIP-IQA ↑ | 0.4855 | 0.2450 | **0.4966** |
| DOVER ↑ | 0.3996 | 0.0231 | **0.4333** |

*(b)* Weight quantization algorithms.

| Method | No QAO | With QAO |
|---|---|---|
| PSNR ↑ | 23.23 | **23.80** |
| DISTS ↓ | 0.2582 | **0.2348** |
| CLIP-IQA ↑ | 0.4650 | **0.4966** |
| DOVER ↑ | 0.4162 | **0.4333** |

*(c)* Layer sensitivity estimation scheme.

| Sensitivity estimation levels | Uniform | | Simplified | | Ours |
|---|---|---|---|---|---|
| | Fully Optimized | Lightly Adapted | Frozen + Lightly Adapted | Frozen + Fully Optimized | Frozen + Lightly Adapted + Fully optimized |
| PSNR↑ | 23.43 | 23.74 | 23.75 | 23.52 | **23.80** |
| DISTS↓ | 0.2424 | 0.2351 | 0.2349 | 0.2395 | **0.2348** |
| CLIP-IQA↑ | 0.4801 | 0.4966 | 0.4901 | 0.4848 | **0.4966** |
| DOVER↑ | 0.4211 | 0.4328 | 0.4395 | **0.4571** | 0.4333 |

*(d)* Multiple sizes of calibration dataset.

| Dataset size | 50 | 100 | 200 |
|---|---|---|---|
| $\delta_{1,low}$ ($\times 10^{-5}$) | 3.03 | 2.50 | 2.47 |
| $\delta_{1,high}$ ($\times 10^{-3}$) | 1.88 | 1.65 | 1.63 |
| $\delta_{2,low}$ | 0.069 | 0.068 | 0.066 |
| $\delta_{2,high}$ | 0.089 | 0.080 | 0.079 |

*(e)* Multiple choices of random seeds.

| Random seed value | 42 | 172 | 3856 |
|---|---|---|---|
| $\delta_{1,low}$ ($\times 10^{-5}$) | 3.03 | 2.96 | 2.64 |
| $\delta_{1,high}$ ($\times 10^{-3}$) | 1.88 | 1.70 | 1.64 |
| $\delta_{2,low}$ | 0.069 | 0.068 | 0.070 |
| $\delta_{2,high}$ | 0.089 | 0.087 | 0.087 |

locate varying numbers of training iterations based on different layer sensitivity estimation schemes: (1) a uniform scheme where all trainable layers are treated equally as either lightly-adapted or fully-optimized; (2) a simplified scheme where unfrozen layers are treated identically; and (3) our proposed scheme featuring a three-tier classification. Results in Table 3c confirm that the proposed classification strategy achieves better performance, and it is significant for reasonably defining the importance and sensitivity of different types of layers in the diffusion transformer module, especially in a one-step inference scenario.

Moreover, we conduct ablation experiments on the calibration sample size and random seed. The results indicate that, regardless of the number of calibration samples or the random seed used, layer sensitivity consistently reveals three distinct clusters with clear gaps. Although the specific numerical boundaries of each cluster may drift slightly with the setting, the qualitative grouping remains consistent. As shown in Table 3d, as the number of calibration samples increases, the lower bound of $\delta_2$ changes slightly, yet the three-cluster structure remains stable, and the cluster assignment of layers stays unchanged. The results in Table 3e further confirm that the threshold ranges and cluster assignments obtained under different random seeds are highly consistent, indicating that the sensitivity measure is robust to random fluctuations. Overall, a larger calibration set helps to refine the numerical boundaries of layer sensitivity, but even with only 50 samples, the clustering structure is stable enough to reveal different layer sensitivities, providing a solid experimental basis for the robust application of VOLTS in small-sample scenarios.

**Quantization Aware Optimization (QAO).** We compare model performance with and without the proposed quantization-aware alternating optimization. It should be noted that omitting QAO is equivalent to treating every trainable layer identically as frozen within VOLTS in practice. As demonstrated in Table 3b, the QAO strategy leads to clearly better results, which proves that it is essential to jointly optimize the quantized and high-precision branches and get better output quality. The iterative approach applied in QAO has a minor weight quantization error.

## 5. Conclusion

We introduce LSGQuant in this article, an approach for video super-resolution tasks relying on one-step diffusion and its low-bit quantization. To effectively handle video tokens, we introduce a Dynamic-Range Adaptive Quantizer (DRAQ), which significantly mitigates activation quantization errors. Furthermore, we design a Variance-Oriented Layer Training Strategy (VOLTS) and a Quantization-Aware Alternating Optimization (QAO) scheme under strict one-step inference constraints to better optimize diffusion transformers. Extensive testing results prove that LSGQuant achieves efficient and high-quality output video compared to full-precision baselines and has practical value.

## Acknowledgement

This work was supported by National Natural Science Foundation of China (62501386, 625B2116, 2024YFC3017100, YDZX20253100004004, 62302299), Huawei Explore X funds, CCF-Tencent Rhino-Bird Open Research Fund, and CAAI-Tencent Rhino-Bird Open Research Fund. This work is also sponsored by AI Hundred Schools Program and is carried out using the Ascend AI technology stack.

## Impact Statement

This paper presents work whose goal is to advance the field of Machine Learning. There are many potential societal consequences of our work, none which we feel must be specifically highlighted here.

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
