# LSGQuant: Layer-Sensitivity Guided Quantization for One-Step Diffusion Real-World Video Super-Resolution
## *Supplementary Materials*

## Overview

In this supplementary material, we provide more details and results including following contents:

- **Implementation Details**: Basic settings of comparative methods and specific designs of our LSGQuant.

- **Additional Experiments**: Experiments conducted on additional synthetic dataset SPMCS (Yi et al., 2019), YouHQ40 (Zhou et al., 2024) and real-world dataset RealVSR (Yang et al., 2021).

- **More Visual Results**: More quantitative results, including visual quality and temporal consistency.

- **Limitations and Future Work**: Analysis of current limitations and suggesstions for future improvements.

## Implementation Details

In this section, we provide additional implementation details for both comparative methods and our proposed approach.

**Implementation of comparative methods**. Due to the lack of existing quantization methods specifically designed for one-step diffusion models in video super-resolution tasks, we adapt representative approaches from related research domains. These include low-bit quantization methods originally developed for multi-step diffusion models as well as large language models (LLMs).

We re-implement ViDiT-Q (Zhao et al., 2025) and SVDQuant (Li et al., 2025) based on their official code releases. The ViDiT-Q codebase provides implementations of several representative quantization techniques, including MinMax (Jacob et al., 2018), SmoothQuant (Xiao et al., 2023), and QuaRot (Ashkboos et al., 2024), which are directly adopted in our experiments. Following the quantization settings of ViDiT-Q, all comparative methods employ an asymmetric static weight quantizer together with a symmetric dynamic activation quantizer. For pre-scaling in SmoothQuant and ViDiT-Q, the scaling factor $\alpha$ is set to 0.5. In SVDQuant, the rank of the high-precision branch is set to $r = 32$, which is consistent with the setting used in our LSGQuant method to ensure a fair comparison.

**More details of LSGQuant**. The core configurations of our proposed components, including DRAQ, VOLTS, and QAO, are described in the main paper. Here, we provide several additional implementation details. During the post-training quantization (PTQ) procedure in VOLTS, after completing the layer sensitivity estimation in Step 2, we sort all candidate sensitivity values to determine the thresholds for the three-class classification. In the iterative optimization process of QAO, we adopt an error-based early stopping criterion to ensure both stability and efficiency. Specifically, we compare the quantization errors between two consecutive iterations; if no further error reduction is observed for more than 10 consecutive iterations, the optimization process is terminated.

## Additional Experiments

In addition to the UDM10 (Tao et al., 2017) and REDS30 (Nah et al., 2019), and real-world dataset MVSR4x (Wang et al., 2023) discussed in the main paper, we conducted further experiments on the synthetic dataset SPMCS (Yi et al., 2019), YouHQ40 (Zhou et al., 2024) and real-world dataset RealVSR (Yang et al., 2021). For all synthetic dataset, we apply a same degradation pipeline for. As reported in Tab 1, our method outperforms existing approaches on most metrics under 4-bit quantization. In 6-bit scenarios, no-referenced based IQA metrics and VQA metrics seems lower. The synthetic dataset comprises a wide variety of real-world degradations and stylistically diverse video content, whose distribution deviates from the training data of typical no-reference quality assessment models. Consequently, no-reference metrics become more sensitive to subtle texture differences and stylistic shifts, thereby amplifying fluctuations in perceptual scores. Nevertheless, the consistent improvement of full-reference metrics demonstrates that our method retains an advantage in restoring authentic structural details and suppressing distortions induced by low-bit quantization. Considering the experimental results across both 4-bit and 6-bit settings, as well as

on synthetic and real-world datasets, we contend that these local discrepancies in certain metrics do not undermine the overall conclusion: our proposed LSGQuant delivers well performance for the task of low-bit video super-resolution.

## More Visual Results

In this section, we present more visual results on both main datasets in article and supplementary datasets, including comparisons of visual quality and temporal consistency.

**Visual Comparison**. We provide additional visual comparison results. First, we compare all datasets outcomes under 4-bit quantization, as illustrated in Figs **??**, respectively. Under the 4-bit quantization setting, our LSGQuant significantly outperforms competing method in either synthetic or real-world scenarios. Under the 6-bit quantization setting shown in Fig 4, recent state-of-the-art quantization methods (*i.e.* ViDiT-Q (Zhao et al., 2025) and SVDQuant (Li et al., 2025)) perform reasonably well, but still exhibit shortcomings in certain complex scenarios. In contrast, our method effectively addresses these issues and achieves performance close to the full-precision model. Furthermore, we compare LSGQuant with ViDiT-Q and SVDQuant on twin scenarios at both 6- bit and 4-bit settings, as shown in Fig 5.

**Temporal Consistency**. We provide temporal profiles on other different datasets, as shown in Fig (**?**). Quantization could cause inconsistent errors across frames. The temporal profiles of comparison methods reveal a distinctly segmented appearance across frames, indicating discontinuities between frames. In contrast, QuantVSR demonstrates smoother temporal profiles across various scenarios.

## Limitations and Future Work

In this work, we propose LSGQuant, an effective low-bit quantization method for real-world video super-resolution based on one-step diffusion models. Similar to existing diffusion-based quantization approaches, our framework primarily focuses on the Diffusion Transformer (DiT) component in WAN2.1-1.3B (Wan et al., 2025), as it accounts for the majority of computational and memory overhead and constitutes the main performance bottleneck.

However, we observe that the Variational Autoencoder (VAE) module also consumes a non-negligible amount of GPU memory, despite its relatively low inference latency. This indicates that further memory reduction could be achieved by extending quantization to all model components. Therefore, future work will explore unified low-bit quantization strategies jointly optimize them.

Overall, LSGQuant achieves favorable trade-offs between video super-resolution performance and model compression under low-bit settings, demonstrating its potential value for

both academic research and industrial applications. We do not foresee any negative societal or industrial impacts arising from the deployment of this method on resource-constrained edge devices.

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

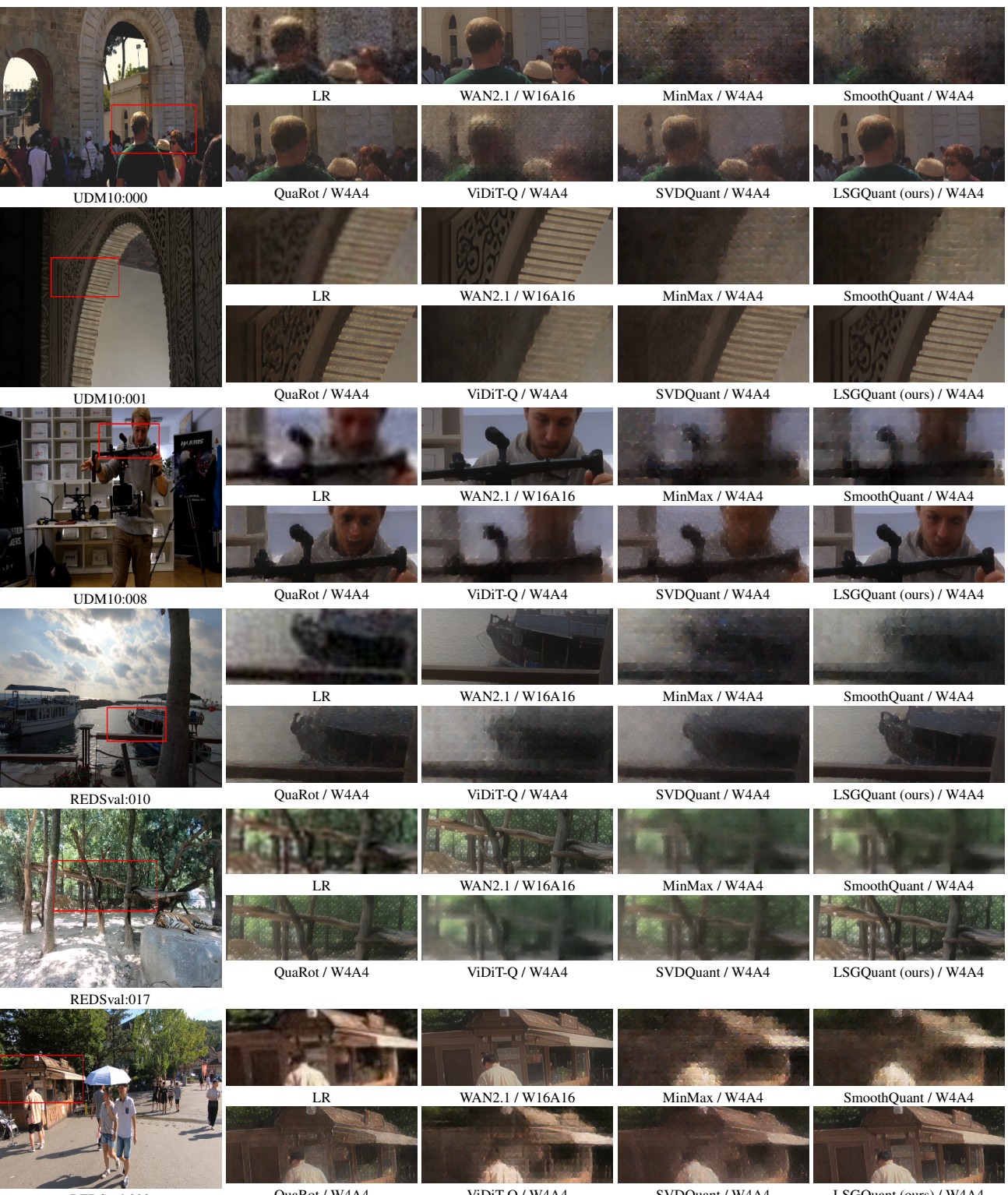

*Figure 1.* More visual comparisons on various datasets at 4-bit quantization.

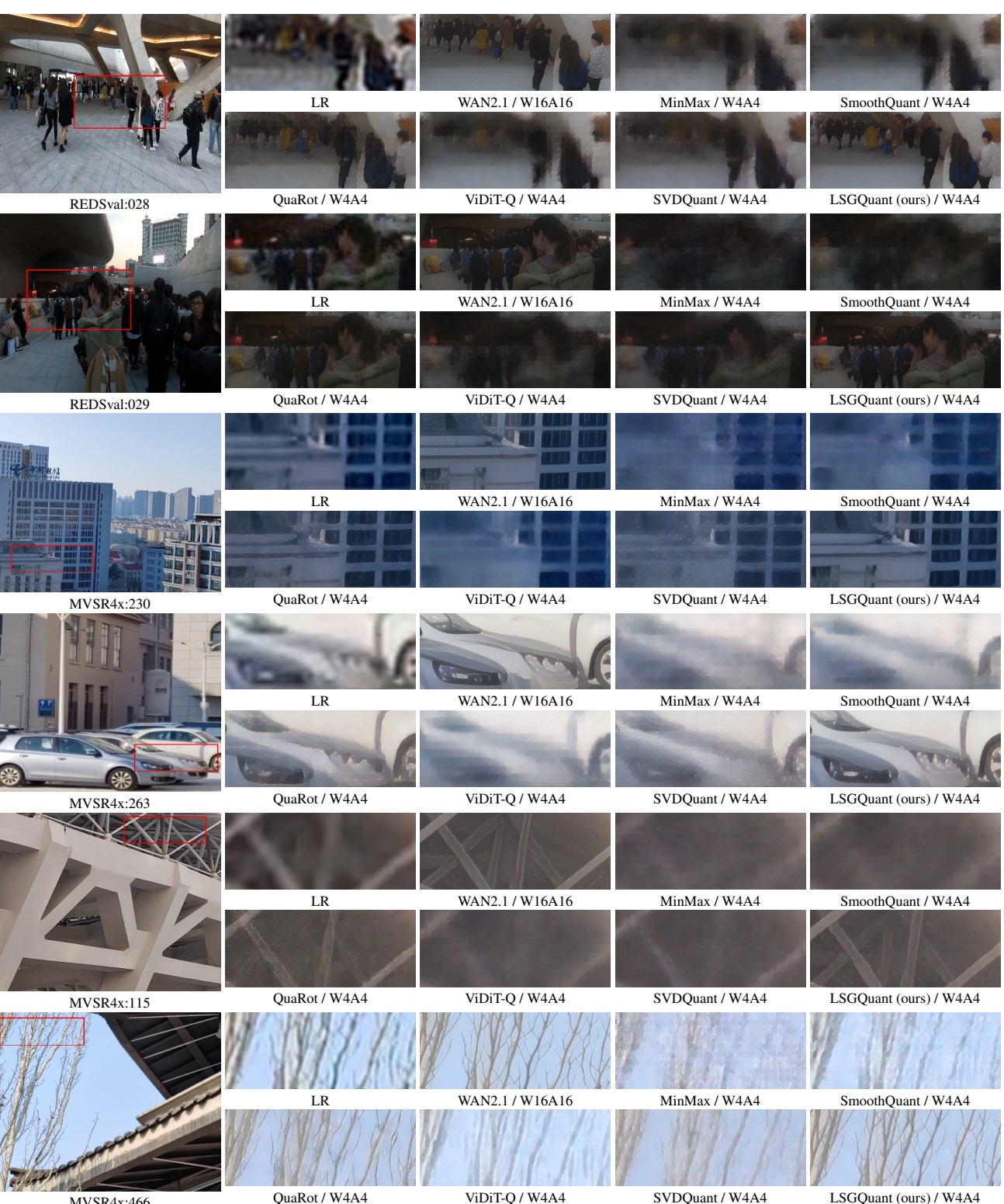

*Figure 2.* More visual comparisons on various datasets at 4-bit quantization.

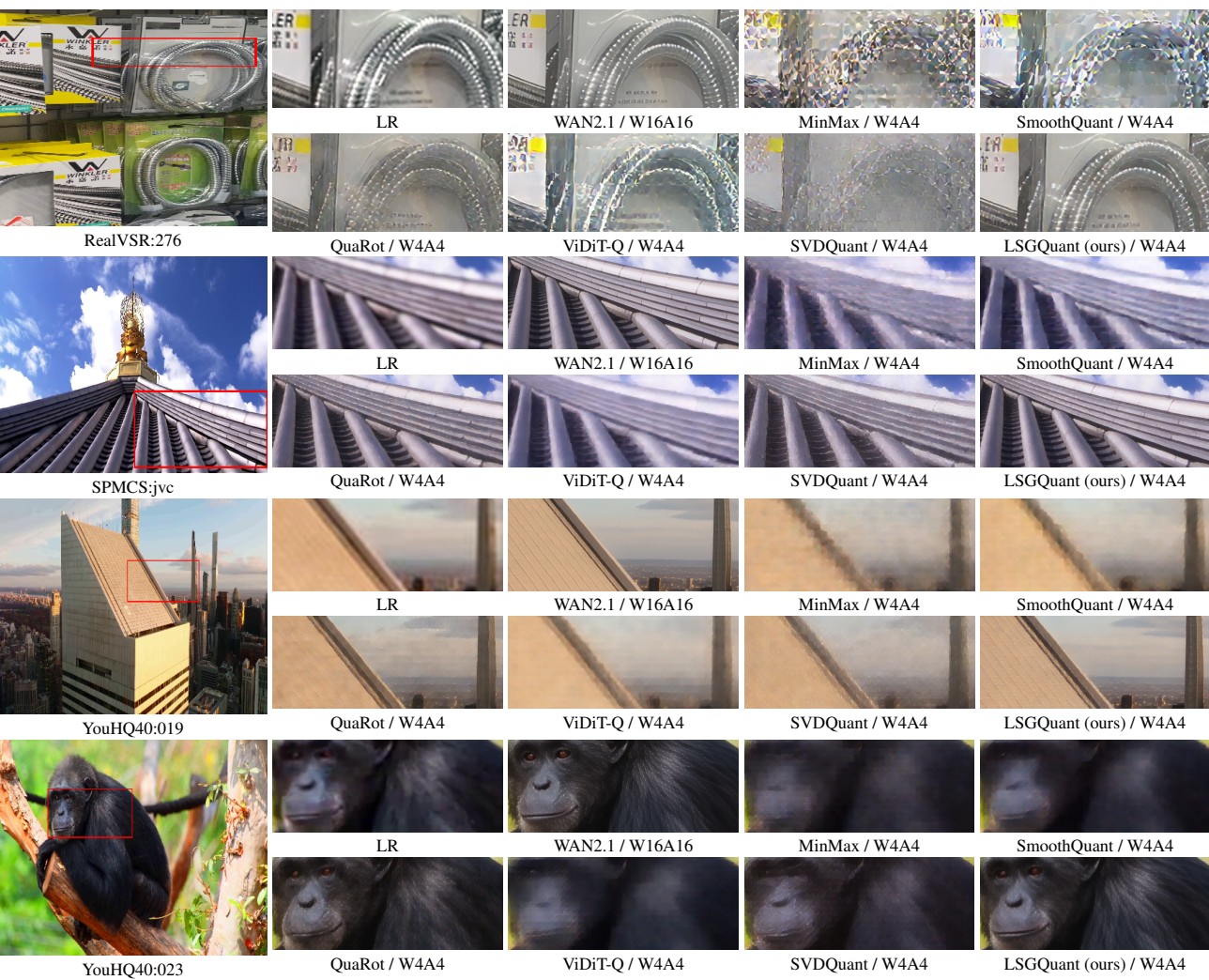

*Figure 3.* All possible results.

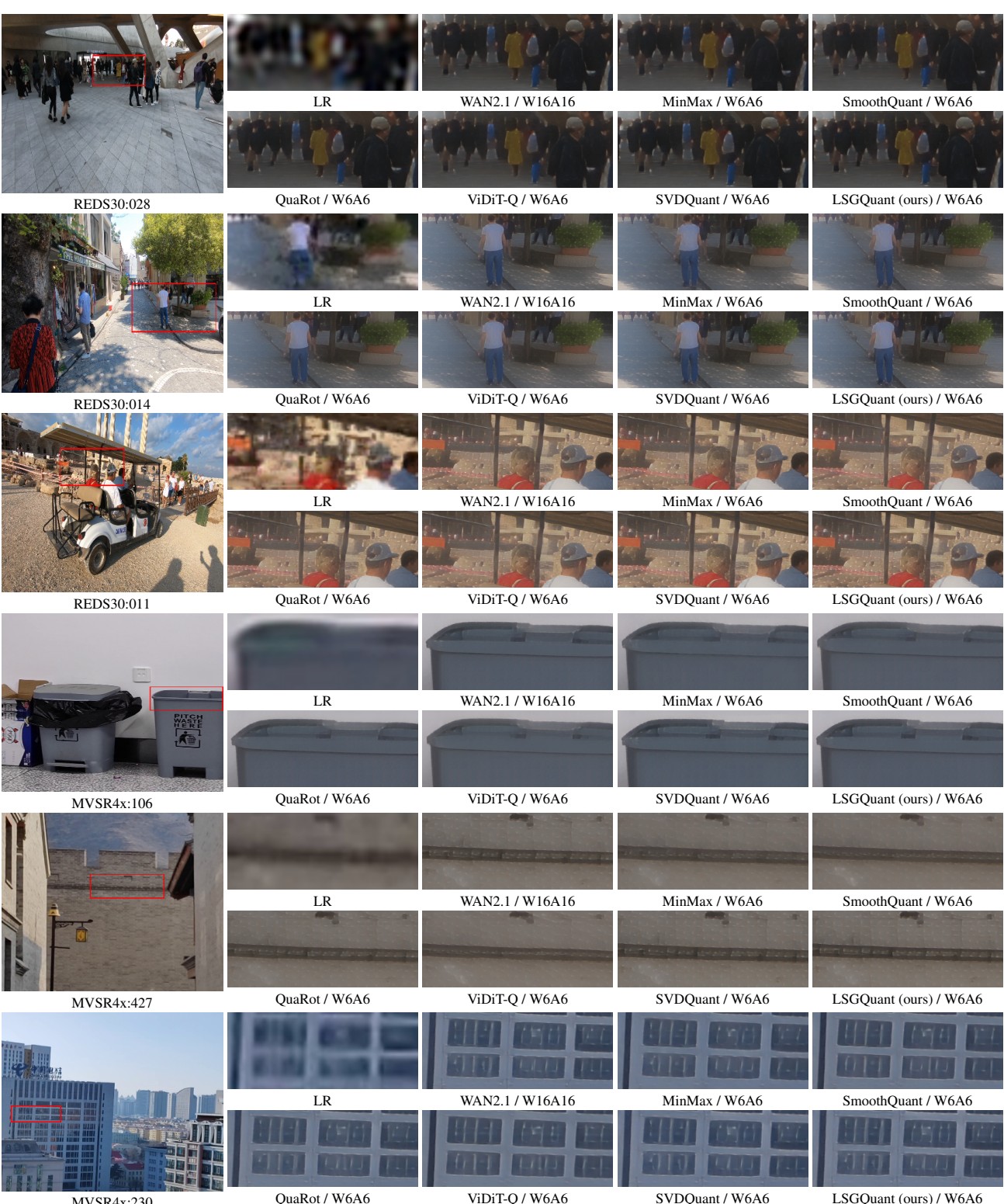

*Figure 4.* Visual comparison on REDS30 (Nah et al., 2019)) and MVSR4x (Wang et al., 2023) at 6-bit quantization.

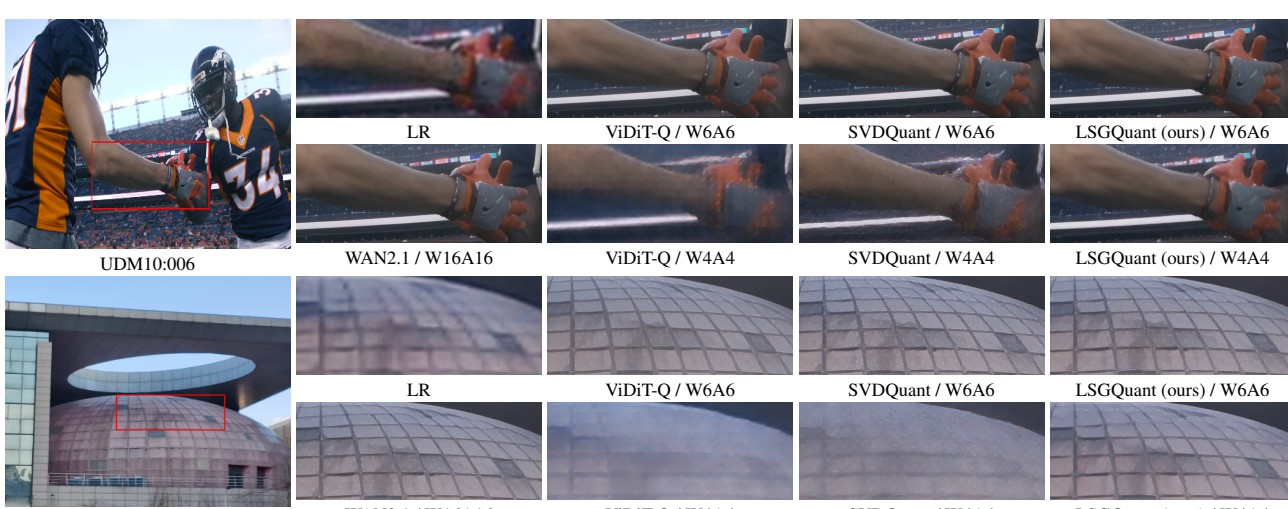

*Figure 5.* Visual comparisons with recent leading diffusion quantization methods i.e., ViDit-Q (Zhao et al., 2025) and SVDQuant (Li et al., 2025) ) at 6-bit / 4-bit quantization. Our method performance exceeds leading quantization methods on both synthetic and real-world scenario.

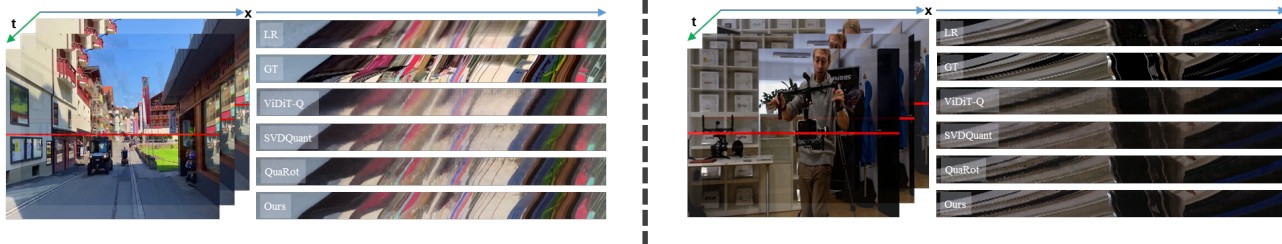

*Figure 6.* Visual comparisons on temporal consistency with diffusion quantization methods. Our method performs well temporal continuity.

*Table 1.* Quantitative results on extra synthetic SPMCS (Yi et al., 2019), YouHQ40 (Zhou et al., 2024) and real-world dataset RealVSR (Yang et al., 2021). Red and Blue texts represent the best and second best scores, respectively.

| Dataset | Bit | Method | PSNR↑ | SSIM↑ | LPIPS↓ | DISTS↓ | CLIP-IQA↑ | MUSIQ↑ | MANIQA↑ | DOVER↑ | $E^*_{warp}$↓ |
|---|---|---|---|---|---|---|---|---|---|---|---|
| SPMCS | W16A16 | WAN (Wan et al., 2025) | 21.93 | 0.5670 | 0.2791 | 0.2518 | 0.4584 | 68.95 | 0.3819 | 0.4394 | 0.58 |
| | W6A6 | MinMax (Jacob et al., 2018) | 21.62 | 0.5617 | 0.2907 | 0.2625 | 0.4589 | 69.00 | 0.3776 | 0.4131 | 0.60 |
| | | SmoothQuant (Xiao et al., 2023) | 21.46 | 0.5567 | 0.2922 | 0.2626 | 0.4632 | 69.66 | 0.3833 | 0.4461 | 0.56 |
| | | QuaRot (Ashkboos et al., 2024) | 21.86 | 0.5663 | 0.2822 | 0.2564 | 0.4552 | 68.76 | 0.3811 | 0.4251 | 0.57 |
| | | ViDiT-Q (Zhao et al., 2025) | 21.48 | 0.5562 | 0.2913 | 0.2615 | 0.4630 | 69.98 | 0.3824 | 0.4555 | 0.56 |
| | | SVDQuant (Li et al., 2025) | 21.91 | 0.5662 | 0.2806 | 0.2551 | 0.4561 | 68.60 | 0.3777 | 0.4222 | 0.59 |
| | | LSGQuant(ours) | 21.93 | 0.5664 | 0.2812 | 0.2514 | 0.4625 | 68.81 | 0.3793 | 0.4314 | 0.58 |
| | W4A4 | MinMax (Jacob et al., 2018) | 19.31 | 0.4736 | 0.5957 | 0.4439 | 0.2534 | 29.86 | 0.2015 | 0.0892 | 1.85 |
| | | SmoothQuant (Xiao et al., 2023) | 19.76 | 0.4909 | 0.5641 | 0.4178 | 0.2386 | 29.76 | 0.1912 | 0.0875 | 1.64 |
| | | QuaRot (Ashkboos et al., 2024) | 20.88 | 0.5391 | 0.3546 | 0.3075 | 0.4293 | 62.37 | 0.3110 | 0.3177 | 0.94 |
| | | ViDiT-Q (Zhao et al., 2025) | 20.31 | 0.5161 | 0.5407 | 0.4005 | 0.2456 | 29.07 | 0.1868 | 0.0942 | 0.88 |
| | | SVDQuant (Li et al., 2025) | 20.37 | 0.5019 | 0.4673 | 0.4014 | 0.4708 | 49.77 | 0.2897 | 0.2021 | 0.90 |
| | | LSGQuant(ours) | 21.97 | 0.5687 | 0.3069 | 0.2678 | 0.4368 | 64.65 | 0.3394 | 0.3586 | 0.66 |
| YouHQ40 | W16A16 | WAN (Wan et al., 2025) | 22.01 | 0.5967 | 0.3478 | 0.2357 | 0.5663 | 66.73 | 0.3777 | 0.6864 | 1.54 |
| | W6A6 | MinMax (Jacob et al., 2018) | 21.49 | 0.5922 | 0.3630 | 0.2452 | 0.5603 | 65.90 | 0.3671 | 0.6505 | 1.36 |
| | | SmoothQuant (Xiao et al., 2023) | 21.40 | 0.5870 | 0.3666 | 0.2477 | 0.5718 | 66.83 | 0.3759 | 0.6709 | 1.40 |
| | | QuaRot (Ashkboos et al., 2024) | 21.92 | 0.5971 | 0.3520 | 0.2386 | 0.5603 | 66.11 | 0.3742 | 0.6710 | 1.49 |
| | | ViDiT-Q (Zhao et al., 2025) | 21.50 | 0.5891 | 0.3680 | 0.2465 | 0.5742 | 66.70 | 0.3728 | 0.6682 | 0.92 |
| | | SVDQuant (Li et al., 2025) | 21.99 | 0.5968 | 0.3525 | 0.2371 | 0.5666 | 66.15 | 0.3747 | 0.6734 | 1.42 |
| | | LSGQuant(ours) | 21.98 | 0.5949 | 0.3495 | 0.2363 | 0.5716 | 66.75 | 0.3775 | 0.6800 | 1.26 |
| | W4A4 | MinMax (Jacob et al., 2018) | 19.37 | 0.5477 | 0.5809 | 0.4312 | 0.2808 | 23.77 | 0.1981 | 0.0943 | 2.03 |
| | | SmoothQuant (Xiao et al., 2023) | 19.97 | 0.5601 | 0.5496 | 0.3973 | 0.2769 | 25.19 | 0.1949 | 0.1413 | 2.02 |
| | | QuaRot (Ashkboos et al., 2024) | 20.86 | 0.5734 | 0.4258 | 0.2875 | 0.5158 | 55.90 | 0.2967 | 0.4810 | 1.63 |
| | | ViDiT-Q (Zhao et al., 2025) | 20.45 | 0.5784 | 0.5309 | 0.3764 | 0.2847 | 24.35 | 0.1885 | 0.1805 | 1.47 |
| | | SVDQuant (Li et al., 2025) | 20.35 | 0.5631 | 0.5086 | 0.3797 | 0.5402 | 38.85 | 0.2690 | 0.3058 | 1.44 |
| | | LSGQuant(ours) | 22.11 | 0.6040 | 0.3635 | 0.2427 | 0.5510 | 60.85 | 0.3383 | 0.6138 | 1.84 |
| RealVSR | W16A16 | WAN (Wan et al., 2025) | 20.20 | 0.6051 | 0.1921 | 0.1551 | 0.5462 | 75.00 | 0.4488 | 0.6171 | 2.40 |
| | W6A6 | MinMax (Jacob et al., 2018) | 19.69 | 0.5773 | 0.2163 | 0.1670 | 0.5436 | 75.04 | 0.4532 | 0.5903 | 2.34 |
| | | SmoothQuant (Xiao et al., 2023) | 19.68 | 0.5757 | 0.2162 | 0.1680 | 0.5489 | 75.05 | 0.4597 | 0.6095 | 2.32 |
| | | QuaRot (Ashkboos et al., 2024) | 20.09 | 0.5988 | 0.1969 | 0.1581 | 0.5445 | 74.95 | 0.4482 | 0.6158 | 2.37 |
| | | ViDiT-Q (Zhao et al., 2025) | 19.77 | 0.5798 | 0.2127 | 0.1662 | 0.5462 | 75.05 | 0.4580 | 0.6127 | 2.34 |
| | | SVDQuant (Li et al., 2025) | 20.19 | 0.6033 | 0.1937 | 0.1530 | 0.5525 | 74.94 | 0.4448 | 0.6122 | 2.40 |
| | | LSGQuant(ours) | 20.15 | 0.6024 | 0.1930 | 0.1550 | 0.5465 | 74.85 | 0.4439 | 0.6133 | 2.37 |
| | W4A4 | MinMax (Jacob et al., 2018) | 16.26 | 0.3962 | 0.4475 | 0.3040 | 0.5357 | 61.67 | 0.3416 | 0.2501 | 18.85 |
| | | SmoothQuant (Xiao et al., 2023) | 16.44 | 0.4293 | 0.3844 | 0.2689 | 0.5693 | 65.13 | 0.3935 | 0.2786 | 19.52 |
| | | QuaRot (Ashkboos et al., 2024) | 18.80 | 0.5248 | 0.2676 | 0.2013 | 0.5639 | 72.52 | 0.3872 | 0.5182 | 2.97 |
| | | ViDiT-Q (Zhao et al., 2025) | 17.59 | 0.5092 | 0.2858 | 0.1988 | 0.5871 | 68.90 | 0.4199 | 0.4459 | 10.84 |
| | | SVDQuant (Li et al., 2025) | 17.89 | 0.4344 | 0.4026 | 0.2889 | 0.5714 | 66.50 | 0.3577 | 0.3428 | 3.31 |
| | | LSGQuant(ours) | 0.08 | 0.5885 | 0.2085 | 0.1632 | 0.5964 | 73.86 | 0.4294 | 0.5609 | 2.87 |