# OpenReview forum: "LSGQuant: Layer-Sensitivity Guided Quantization for One-Step Diffusion Real-World Video Super-Resolution"
_ICML.cc/2026/Conference — ICML 2026 regular_

### Official Review · Reviewer_DNzn · 2026-03-03

**Soundness:** 3
**Presentation:** 2
**Significance:** 2
**Originality:** 2
**Overall Recommendation:** 4
**Confidence:** 3

**Summary:**

The paper analyzes the sources of quantization error in one-step video diffusion models.
They propose an inference strategy that for each layer adds a full-precision low-rank approximation and a quantized low-precision high-rank residual. Quantization ranges are adjusted per-channel and per-token, low-rank approximations are chosen to minimizes quantization errors, and calibration is focused mainly on some important layers.

**Compliance With Llm Reviewing Policy:**

Affirmed.

**Final Justification:**

Additional experiments and clarifications were clearly reasoned. The paper presents a well-engineered method that achieves a reasonable improvement over previous work.

**Key Questions For Authors:**

- Why can't $s_i$ be computed as in eq. (2)? This is a generalization of the authors proposed scales. The authors argue that calibrating $\alpha$ is not possible with a single-step model, but shouldn't this be possible to tune in the calibration step?
- Figure 3 shows that X is immediately de-quantized to be used in the next layer. Shouldn't this happen after multiplication with Q(W) so the matrix multiplication can be done in low-precision?
- In 3.4. why is the sensitivity metric chosen as the variance of the mean activation. Wouldn't it be more fitting to also account for the variance of the activations?
- Which layers of the model end up being frozen, lightly adapted, fully optimized?

**Limitations:**

Not discussed in paper. An analysis of why their method fails to be state-of-the-art for 6-bit would be welcomed.

**Strengths And Weaknesses:**

Strengths
- Decent analysis of sources of quantization error
- Great results on fast 4-bit quantized video super-resolution

Weaknesses
- Paper could be more concise (e.g. reduce redundancies between introduction and contributions list). Many typos ('statiscics', 'preocess', ...) and other writing mistakes (missing 'the's, incorrect sentence structure) harm readability. I suggest the author's make use of appropriate tools to find and fix these.
- Some captions (e.g. Fig 1 and 2) were minimal and didn't fully explain their figure in a self-contained way.
- Some arguments are hard to follow or seem to lack evidence, e.g.: 'As illustrated in Fig. 4(a), a small number of channels contain extreme outlier values' -> the figure does not show values by channel, so they could as easily be found in all channels.
- The quantization $Q_A$ in eq. (4) is poorly explained and motivated. The major difference to the baseline in eq. (2) seems to be an additional per-token scaling $d$. Both eq. (2) use per-channel scaling so this is not the main contribution contrary to claim. Furthermore $Q_W$ is never discussed. Finally, as-is, eq. (4) doubly scale entries. As an example let $C=2, N=2, X = [(10, 1), (1, 10)]$. Then $s_1=s_2=d_1=d_2=10$, and $d^{-1}Xs^{-1} = [(0.1, 0.01), (0.1, 0.01)]$. Maybe the authors intended to set $d_i = \max_C |\tilde{X}_{i, C}| / s_i$ to avoid this?

Then
- The paper claim they outperform other methods that are quantized to k-bits but having a full-precision low-rank component makes this a spurious comparison unless the baselines also have such a component. Especially for k=4, it doesn't seem unreasonable to assume that this is where most of the model's capacity comes from. Maybe the others could compare at similar wall-clock inference times?
- The paper claims that one of their main contributions is layer-specific treatment. They claim '[other methods] assume that
all network layers contribute equally to output quality' but quoting from the VIDIT-Q paper (Zhao et al, 2025): 'This suggests that layers have varying quantization sensitivity, and quantization can be ”bottlenecked” by certain highly sensitive layers'.

---

> ### Author Rebuttal · Authors · 2026-03-31
>
> # Response to Reviewer DNzn
> `Q3-1` Paper could be more concise. Many typos and other writing mistakes harm readability.
>
> `A3-1` Thank you for carefully pointing out the typo and notation issues. We acknowledge paper requires closer attention to detail. We have thoroughly revised the camera‑ready version to correct all typos and improve notations.
>
> `Q3-2` Some captions (e.g., Fig.1/2) were minimal and didn't fully explain their figure in a self-contained way.
>
> `A3-2` Thank you for the comment.
>
> - Fig.1 compares FLOPs, params, and visual quality between bfloat16 and 4-bit methods.
> - Fig.2 shows a visual comparison on REDS30 sample, including a radar chart for performance visualization.
>
> `Q3-3` Some arguments are hard to follow or seem to lack evidence. E.g., Fig.4(a) does not show values by channel.
>
> `A3-3` Thank you for the comment. We intend to highlight that quantization error doesn't depend on channel distribution of outliers and that outliers affect overall dynamic range in minmax method.
>
> `Q3-4` $Q_A$ in Eq.(4) is poorly explained. Its main difference from Eq.(2) appears to be per‑token scaling $d$; $Q_W$ is not discussed; Eq.(4) doubly scaled entries (e.g., $X=[(10,1),(1,10)]$).
>
> `A3-4` Thank you for the comment.
> 1. Our method differs from Eq.(2) in two ways.
> - We use a Hadamard transform instead of a learned smooth mask, making it data‑independent and reducing training costs.
> - Per‑token scaling is applied after transformation, improving quantization robustness.
> 2. Weight quantization is handled jointly by the overall architecture with VOLTS and QAO.
> 3. Eq.(4) is executed sequentially, not as simultaneous scaling. Using the example $X=[(10,1),(1,10)]$:
> - $s_1=s_2=10, \tilde{X}=[(1,0.1),(0.1,1)]$
> - $d_1=d_2=1$. $d$ is generally within $[0,1]$. It equals 1 due to chosen value, which is similar to Eq.(2).
> - Under 4-bit $X_{int}=[(7,1),(1,7)],X_{dequant}=[(10,1.42),(1.42,10)]$.
>
> `Q3-5` The paper claims to outperform other k‑bit methods, but using a full‑precision low‑rank component makes the comparison spurious unless baselines also have it. Maybe compare at similar wall‑clock inference times?
>
> `A3-5` Thank you for the comment.
>
> - SVDQuant includes a full‑precision low‑rank branch as common practice. Tab.1 and Fig.1 compare with rank=32 and bits=4, showing our method has slightly lower cost and better performance.
> - On an A6000 with a 32‑frame 180×318 LR video, FP takes 14.46s, SVDQuant 4.79s, LSGQuant 4.76s (3.01× and 3.04× speedup).
>
> `Q3-6` They claim "assume that all network layers contribute equally to output quality", but ViDiT‑Q states "layers have varying quantization sensitivity".
>
> `A3-6` Thank you for the comment. Our contribution differs from ViDiT-Q in these aspects:
> 1. We adopt a simpler setting and achieve better performance with lower overhead, shown in Tab.1 and Fig.1.
> 2. Differences in model architecture and task result in another sensitivity pattern. We report classification for all 300 linear modules in Wan2.1 DiT blocks:
> - **Low - Frozen**: 90 layers, including `k`, `v`, and `o` of all cross-attention modules, which connect fixed embeddings.
> - **Medium - Lightly adapted**: 193 layers, covering most in DiT.
> - **High - Fully Optimized**: 17 layers, mainly self‑attention near beginning/end of DiT blocks.
>
> `Q3-7` Computing $s_i$ in Eq.(2) generalizes the proposed scales. Authors claim $\alpha$ can't be calibrated in a single-step model, but shouldn't this be possible during calibration?
>
> `A3-7` Thank you for the comment. Scaling factors following Eq.(2) remain static during inference. In contrast, our DRAQ uses input‑adaptive scaling that dynamically adjusts to activation distributions, which is especially critical for lack of temporal redundancy. Tab.4(b) shows static approach has a significant performance drop.
>
> `Q3-8` Fig.3 shows that $X$ is immediately de-quantized to be used in the next layer. Shouldn't this happen after multiplication with $H$ so the matrix multiplication can be done in low-precision?
>
> `A3-8` Thank you for the comment. The dequantization design follows practice in ViDiT-Q and QuaRot for fair comparison.
>
> `Q3-9` In section 3.4, why is the sensitivity metric chosen as the variance of the mean activation? Wouldn't it be more fitting to also account for the variance of the activations?
>
> `A3-9` Thank you for the comment. We selected the variance of the mean activation as our sensitivity metric primarily due to computational resource constraints.
>
> `Q3-10` Which layers of the model end up being frozen, lightly adapted, or fully optimized?
>
> `A3-10` Thank you for the comment. Results are shown in `A3-6`.
>
> `Q3-11` An analysis of why their method fails to be state-of-the-art for 6-bit would be welcomed.
>
> `A3-11` Thank you for the comment.
>
> - Under 6‑bit quantization, quantization error is small, and performance gaps narrow. Our advantage is more pronounced at lower bits.
> - Still, our method achieves SOTA on most metrics in REDS30 and approaches full‑precision on others.

---

> > ### Author Rebuttal · Reviewer_DNzn · 2026-03-31
> >
> > I thank the authors for their thorough response that cleared up several of my questions. Still, I will keep my score as I feel some of my concerns are not fully resolved. Specifically the overall unclear presentation of the method and the missing details for the additional experiments remain a concern. For example,
> >
> > "SVDQuant includes a full‑precision low‑rank branch as common practice"
> > - What about the other methods? What are the speed-up's there? Why is this computed on only one video and not the datasets from e.g. Table 1?
> >
> > "Our advantage is more pronounced at lower bits" "our method achieves SOTA on most metrics in REDS30"
> > - But is not SOTA on the other datasets. Overall this seems to be a major limitation, as it shows that this method provides a highly engineered method that works best in a narrow regime.

---

> > > ### Author Response · Authors · 2026-04-04
> > >
> > > # Thanks Reviewer DNzn for approving our work
> > >
> > > Dear Reviewer DNzn,
> > >
> > > Thank you for acknowledging our work. We sincerely appreciate your valuable suggestions and will refine our paper based on your feedback.
> > >
> > > Best regards,
> > >
> > > Authors
> > >
> > > # Further Discussion on Reviewer DNzn
> > >
> > > Dear Reviewer DNzn,
> > >
> > > Thank you for taking the time to review our response and for your thoughtful and constructive comments. We are glad that our rebuttal has addressed most of your concerns. We have carefully considered your acknowledgement and provided detailed responses to your questions:
> > >
> > > 1. "SVDQuant includes a full‑precision low‑rank branch as common practice"
> > >
> > > - We have evaluated our method on **other low-rank based quantization methods** for diffusion models, including **EfficientDM** [**ref1**] and **QuantVSR** [**ref2**], and tested their performance and average speed-up on the UDM10 dataset.
> > > - UDM10 includes ten 32‑frame low-resolution (180×318) videos, which are **consistent** with the test video mentioned in the rebuttal before. We apologize for the lack of this baseline setting in the rebuttal due to character limitations.
> > >
> > > |Metrics|FP|EfficientDM|SVDQuant|QuantVSR|LSGQuant(ours)|
> > > |-|-|-|-|-|-|
> > > |Inference time(s)|14.46|5.15|4.79|4.79|**4.76**|
> > > |SpeedUp|1x|2.81x|3.01x|3.01x|**3.04x**|
> > > |PSNR↑|23.74|20.77|21.92|22.16|**23.80**|
> > > |DISTS↓|0.2080|0.4146|0.3781|0.2681|**0.2348**|
> > > |DOVER↑|0.5494|0.1072|0.1718|0.3762|**0.4333**|
> > > |$E^*_{warp}$↓|0.93|1.64|1.23|1.12|**1.07**|
> > >
> > > - Low-rank-based designs originated from research on quantization and compression for large language models (LLMs). **EfficientDM** pioneered the application of this design in diffusion model quantization. It has become a common design choice in recent works, as low-rank components offer substantial performance gains at a relatively low computational cost.
> > >
> > > - Unlike conventional low‑rank approaches, which usually require equal training on each layer, we **reallocate training resources** across layers using our `VOLTS` and `QAO`.
> > >   - `VOLTS` evaluates the sensitivity of each linear module to input variations and allocates training resources accordingly. We introduce an early stopping mechanism that `frozen` or `lightly adapted` certain components.
> > >   - `QAO` reflects the per‑module mechanism of VOLTS in the number of iterations.
> > >
> > > 2. "Our advantage is more pronounced at lower bits" "our method achieves SOTA on most metrics in REDS30"
> > >
> > > - As noted in our paper and supplementary material, the following table reports where LSGQuant get first or second performance among 6-bit methods on all datasets.
> > >
> > > |Metrics|UDM10|REDS30|MVSR4x|SPMCS|YouHQ40|RealVSR
> > > |-|-|-|-|-|-|-|
> > > |PSNR↑|2nd|2nd|2nd|**1st**|2nd|2nd
> > > |SSIM↑|2nd|2nd|-|**1st**|-|2nd|
> > > |LPIPS↓|2nd|**1st**|**1st**|2nd|**1st**|**1st**|
> > > |DISTS↓|**1st**|**1st**|**1st**|**1st**|**1st**|2nd|
> > > |MUSIQ↑|-|**1st**|**1st**|2nd|2nd|-|
> > > |DOVER↑|-|**1st**|**1st**|-|**1st**|2nd|
> > >
> > > While our method does not achieve SOTA everywhere, it remains competitive across all evaluated settings. Being 2nd on some metrics still means we outperform all 4-bit baselines and most 6-bit competitors.
> > >
> > > - Despite practical engineering efforts, our work also introduces **theoretical innovations**, whose effectiveness is validated by the ablation experiments in Tab.3:
> > >   - `DRAQ` introduces a token‑wise scaling operation, allowing us to suppress outliers from both the token and channel perspectives.
> > >   - `VOLTS` uses input variations as the metric for module sensitivity estimation. It directly determines training iteration budgets with simpler quantization settings and the unified metric.
> > >   - `QAO` derives a training strategy that jointly optimizes a quantized branch and a full‑precision low‑rank branch, with early stopping and dynamically freezing insensitive layers.
> > >
> > > - To further show generalization beyond our original setting, we conduct additional experiments on a representative multi‑step method, **MGLD-VSR** [**ref3**]. The results under 4‑bit on the SPMCS dataset show that LSGQuant has potential applicability to broader scenarios.
> > >
> > > |Metrics|ViDiT-Q|SVDQuant|LSGQuant(ours)|
> > > |-|-|-|-|
> > > |PSNR↑|20.60|21.17|**22.62**|
> > > |DISTS↓|0.2911|0.3126|**0.1807**|
> > > |DOVER↑|0.5294|0.5627|**0.6727**|
> > > |$E^*_{warp}$↓|3.98|2.77|**1.82**|
> > >
> > > We hope this response addresses your concern. Thank you again for your thoughtful feedback and kind support.
> > >
> > > [**ref1**] EfficientDM: Efficient Quantization-Aware Fine-Tuning of Low-Bit Diffusion Models, 2024.
> > >
> > > [**ref2**] QuantVSR: Low-Bit Post-Training Quantization for Real-World Video Super-Resolution, 2026.
> > >
> > > [**ref3**] Motion-Guided Latent Diffusion for Temporally Consistent Real-world Video Super-resolution, 2024.
> > >
> > > Best regards,
> > >
> > > Authors

---

### Official Review · Reviewer_nJ4m · 2026-03-05

**Soundness:** 3
**Presentation:** 3
**Significance:** 2
**Originality:** 2
**Overall Recommendation:** 5
**Confidence:** 4

**Summary:**

This paper proposes LSGQuant, a layer-sensitivity-guided quantization framework designed for single-step diffusion models in practical video super-resolution (VSR). Recognizing that diffusion-based VSR models suffer from substantial computational and memory costs, the authors propose a structured low-bit quantization strategy consisting of three components. First, Dynamic Range Adaptive Quantization (DRAQ) is introduced to better capture the high dynamic range of video token activations. Second, a Variance-Oriented Layer Training Strategy (VOLTS) uses calibration variance to estimate layer sensitivity and reallocates layer-wise training budgets/iterations accordingly. Third, Quantization-Aware Alternating Optimization (QAO) jointly optimizes quantized and high-precision branches to mitigate accuracy degradation. Experimental results indicate that LSGQuant achieves competitive reconstruction quality relative to the full-precision backbone, while reducing precision to low-bit settings and reporting substantial compression in DiT-module-only Params/Ops (e.g., W4A4: Params −70.49%, Ops −71.92%).

**Compliance With Llm Reviewing Policy:**

Affirmed.

**Final Justification:**

I had a similar initial concern regarding novelty. At first glance, the method appeared to be mainly an integration of three relatively standard components, tailored to achieve a reasonably good score in a specific engineering setting. I also did not see additional supplementary material providing stronger mathematical justification during my initial review.

That said, I do not think this necessarily precludes the work from being a meaningful engineering contribution. In many engineering fields, including materials science, new performance gains can still arise from nontrivial combinations or arrangements of existing elements(e.g., different arrangements of carbon atoms can lead to distinct material properties). In that sense, I believe the present work still represents meaningful progress in engineering-oriented computer vision applications.

For this reason, while I still acknowledge the limitation in conceptual novelty, I believe the paper merits consideration based on its practical engineering value.In summary, I tend to accept.

**Key Questions For Authors:**

Q1.	On the theoretical justification of layer sensitivity estimation.
The proposed VOLTS module uses activation variance during calibration as a proxy for layer sensitivity. While the intuition is plausible, the manuscript does not provide a formal analysis explaining why variance is an appropriate estimator of quantization-error propagation in diffusion transformers. Could the authors clarify the theoretical or empirical reasoning behind this choice?
(i) How are the thresholds 𝛿1 and 𝛿2selected, and how sensitive are the final results to these hyperparameters?
(ii) The calibration set consists of 50 videos. How does the estimated sensitivity (and thus layer grouping) vary across different calibration subsets/seeds? Reporting variance or stability statistics would strengthen the claim.
(iii) How does variance-based sensitivity compare to Hessian-based or gradient-based criteria commonly used in mixed-precision quantization, either conceptually or via a small controlled comparison?

Q2.	On the stability and convergence behavior of QAO.
The QAO module adopts an alternating optimization strategy between quantized and high-precision branches. Could the authors provide additional evidence regarding training stability, such as convergence curves or sensitivity to initialization? This would help assess whether the proposed optimization is robust or requires careful tuning.

Q3.	On generalization beyond single-step diffusion VSR.
The method is validated primarily on single-step diffusion-based VSR. Could the authors comment on whether LSGQuant generalizes to multi-step diffusion models or other transformer-based generative tasks? Clarifying the scope and limitations would help position the contribution more precisely.

**Limitations:**

While the paper focuses on efficiency improvements for diffusion-based VSR, it would benefit from a clearer discussion of potential failure cases, generalization limits, and broader deployment implications.

**Strengths And Weaknesses:**

This work addresses low-bit deployment for single-step diffusion-based video super-resolution (VSR), a well-motivated and practically relevant problem. The proposed framework is clearly structured, with three main modules coherently organized around the central objective. The experimental evaluation is fairly comprehensive, including both visual comparisons and multiple quantitative metrics. The method demonstrates competitive performance under low-bit settings (e.g., 4-bit), maintaining quality close to full-precision models, which indicates potential value for real-world deployment. However, the theoretical grounding appears relatively limited. In particular, the core module, VOLTS, relies on layer sensitivity estimation without providing sufficient formal justification or deeper analysis of its validity. The contribution is therefore stronger from an engineering integration and task-specific optimization perspective, while its theoretical novelty and depth remain comparatively modest.

---

> ### Author Rebuttal · Authors · 2026-03-31
>
> # Response to Reviewer nJ4m
> `Q2-1` The theoretical grounding appears relatively limited. VOLTS relies on layer sensitivity estimation without sufficient formal justification or deeper analysis of its validity.
>
> `A2-1` Thank you for the comment. Input sensitivity in VOLTS correlates with output quality, motivating finer-grained adaptation, which is consistent with prior works. Moreover, ablation on UDM10 below shows promotion. Further sensitivity-based grouping results are in Tab.4(c).
>
> |Method|VOLTS|PSNR↑|DISTS↓|CLIP-IQA↑|DOVER↑|
> |-|-|-|-|-|-|
> |LSGQuant||21.76|0.3854|0.3038|0.2006|
> |LSGQuant|√(ours)|**23.80**|**0.2348**| **0.4966**|**0.4333**|
>
> `Q2-2` On the theoretical justification of layer sensitivity estimation. Could the authors clarify the theoretical or empirical reasoning behind this choice?
> 1. How are the thresholds $\delta_1$ and $\delta_2$ selected, and how sensitive are the final results to these hyperparameters?
> 2. How does the estimated sensitivity (and thus layer grouping) vary across different calibration subsets/seeds?
> 3. How does variance-based sensitivity compare to Hessian/gradient-based criteria commonly used in mixed-precision quantization?
>
> `A2-2` Thank you for the comment. Layers with larger activation variance have wider dynamic range and higher quantization error, requiring better optimization.
> 1. **Threshold selection and sensitivity.** On 300 linear modules in WAN2.1 DiT, sensitivity scores form three natural clusters with clear gaps, and $\delta_1$ and $\delta_2$ can be chosen anywhere within the gaps without affecting the classification.
> - **Low**: 90 layers, including all `k`, `v`, and `o` of cross-attention modules. $\sigma_l^2 \in [0,3.03 \times 10^{-5}]$.
> - **Medium**: 193 layers, covering most modules in DiT. $\sigma_l^2 \in [1.88 \times 10^{-3},0.069]$.
> - **High**: 17 layers, mainly self‑attention modules near the beginning/end of DiT blocks. $\sigma_l^2 \in [0.089,0.369]$.
>
> 2. **Stability across subsets/seeds.** Overall, sensitivity consistently exhibits three clusters with distinct gaps. While exact scopes vary, qualitative grouping remains unchanged. Larger sets help refine boundaries, and 50 samples are sufficient to reveal the structure.
> - **Other Size**: $\delta_2$ lower bound slightly varies. Three distinct clusters remain unchanged.
>
> |Num|$\delta_{1, low}$ ($10^{-5}$)|$\delta_{1, high}$ ($10^{-3}$)|$\delta_{2, low}$|$\delta_{2, high}$|
> |-|-|-|-|-|
> |50(ours)|3.03|1.88|0.069|0.089|
> |100|2.50|1.65|0.068|0.080|
> |200|2.47|1.63|0.066|0.079|
>
> - **Other Seeds**: Both threshold ranges and results are highly consistent.
>
> |Seed|$\delta_{1, low}$ ($10^{-5}$)|$\delta_{1, high}$ ($10^{-3}$)|$\delta_{2, low}$|$\delta_{2, high}$|
> |-|-|-|-|-|
> |42(ours)|3.03|1.88|0.069|0.089|
> |172|2.96|1.70|0.068|0.087|
> |3856|2.64|1.64|0.070|0.087|
>
> 3. **Comparison with Hessian/gradient-based criteria**. We observe that sensitivity distribution lacks clear separability and differs from origin order, because large input variance doesn't imply large loss impact. These metrics are harder to distinguish layers and have computation overhead, while our method yields well-separated sensitivity with smaller cost. We treat all layers with the same sensitivity for corresponding ablations in Tab.4(c).
>
> `Q2-3` On the stability and convergence behavior of QAO. Could the authors provide additional evidence regarding training stability?
>
> `A2-3` Thank you for the comment. We provide convergence curves for diverse layers [0.ffn.0](https://anonymous.4open.science/r/LSGQuant-rebuttal-357F/0_ffn_0.png), [15.cross_attn.o](https://anonymous.4open.science/r/LSGQuant-rebuttal-357F/15_cross_attn_o.png) and [29.self_attn.q](https://anonymous.4open.science/r/LSGQuant-rebuttal-357F/29_self_attn_q.png). QAO's alternating mechanism is insensitive to initialization.
>
> `Q2-4` On generalization beyond single-step diffusion VSR. Could the authors comment on whether LSGQuant generalizes to multi-step diffusion models or other transformer-based generative tasks?
>
> `A2-4` Thank you for the comment. We conduct additional experiments on a representative multi‑step method **MGLD-VSR** [**ref1**]. The results under 4‑bit on SPMCS show that LSGQuant has potential applicability for multi-step models.
>
> |Metrics|ViDiT-Q|SVDQuant|LSGQuant(ours)|
> |-|-|-|-|
> |PSNR↑|20.60|21.17|**22.62**|
> |DISTS↓|0.2911|0.3126|**0.1807**|
> |DOVER↑|0.5294|0.5627|**0.6727**|
> |$E^*_{warp}$↓|3.98|2.77|**1.82**|
>
> [**ref1**] Motion-Guided Latent Diffusion for Temporally Consistent Real-world Video Super-resolution, 2024.
>
> `Q2-5` The paper would benefit from a clearer discussion of potential failure cases, generalization limits, and broader deployment implications.
>
> `A2-5` Thank you for the comment. We discuss limitation aspects in the supplementary material:
>
> - Our method still struggles with extreme scenarios like severe degradation or small subjects, whose reconstruction quality noticeably degrades.
> - VAE also consumes non‑negligible GPU memory despite low latency.

---

> > ### Author Rebuttal · Reviewer_nJ4m · 2026-04-01
> >
> > Thank you for your response.
> >
> > I could not locate Table 4(a) or (c) in the manuscript (the last table is Table 3)
> > We hope that these contents (including supplementary materials) will be added in subsequent revisions.
> >
> > Overall, the response has largely addressed the main concerns, and we acknowledge the engineering value of the work. The overall assessment is therefore improved accordingly.

---

> > > ### Author Response · Authors · 2026-04-01
> > >
> > > # Thanks Reviewer nJ4m for approving our work
> > >
> > > Dear Reviewer nJ4m,
> > >
> > > Thank you for your response. We are glad that our response addressed your main concerns.
> > >
> > > We apologize for the confusion caused by the incorrect table numbering in our rebuttal. The referenced Table 4(a), (b) and (c) actually correspond to Table 3(a) (DRAQ's ablation study), (b) (QAO's ablation study) and (c) (VOLTS' sensitivity-based grouping results) in the manuscript. We thank the reviewer for pointing this out.
> > >
> > > Best regards,
> > >
> > > Authors

---

### Official Review · Reviewer_sphH · 2026-03-11

**Soundness:** 3
**Presentation:** 2
**Significance:** 3
**Originality:** 2
**Overall Recommendation:** 4
**Confidence:** 3

**Summary:**

This paper introduces LSGQuant, which is the first work to explore low-bit post-training quantization for one-step diffusion models in real-world Video Super-Resolution (VSR). The framework targets Diffusion Transformer (DiT) architectures, utilizing WAN2.1 as its full-precision backbone. To address the challenges posed by the high dynamic range of input latent and diverse layer behaviors in one-step inference, LSGQuant combines three components: a Dynamic-Range Adaptive Activation Quantizer (DRAQ), a Variance-Oriented Layer Training Strategy (VOLTS) that allocates different optimization budgets across layers, and a Quantization-Aware Alternating Optimization (QAO) procedure for the low-rank high-precision branch and quantized branch.
Experiments across synthetic (UDM10, REDS30) and real-world (MVSR4x) datasets demonstrate the efficacy of the proposed framework. Under 4-bit and 6-bit settings, LSGQuant substantially reduces parameter counts and computational operations, while maintaining generation quality comparable to the full-precision model and achieving state-of-the-art quantitative performance.

**Compliance With Llm Reviewing Policy:**

Affirmed.

**Final Justification:**

I thank the authors for their rebuttal. Although the response regarding the novelty did not completely convince me, I have read the other reviewers' comments and reconsidered the overall contribution of the paper. Overall, I have decided to raise my score to a Weak Accept.

**Key Questions For Authors:**

1. On Page 4, Equation (3), should the quantized branch use the residual weight (W_R) rather than (W)?
2. For VOLTS, what is the precise scalar statistic used to compute (sigma_l^2) in Equations (5) and (6)? Also, how many layers fall into each of the three buckets for the reported experiments, and how sensitive are the results to (delta_1) and (delta_2)?
3. In Table 1, were all baselines reimplemented or adapted under the same calibration dataset, optimization budget, and architectural compensation capacity? If not, can you clarify the exact fairness protocol?

**Limitations:**

The authors have discussed the limitations.

**Strengths And Weaknesses:**

Strengths:
1. **Practical value and novelty**:
The paper tackles a highly practical and timely problem: compressing one-step diffusion models for real-world Video Super-Resolution (VSR) to facilitate deployment on resource-constrained hardware.
2. **Sound technical design and modularity**:
Three core modules in LSGQuant are well-designed to target specific failure modes of low-bit quantization. The overall algorithmic route is clearly depicted in Figure 3, which successfully makes the operational flow easy to grasp.
3. **State-of-the-Art results and good presentation**:
The empirical results presented in Table 1 are highly competitive, particularly under the aggressive 4-bit (W4A4) setting. The paper is well-presented, and the technical motivations are strongly supported by clear visual evidence (Figure 4).

Weaknesses:
1. **Limited Architectural Novelty**:
While LSGQuant is an effective engineering solution tailored for one-step VSR, the core algorithmic innovation is limited. The three primary modules (DRAQ, VOLTS and QAO) are essentially a careful, task-specific synthesis of existing techniques.
DRAQ calculates the absolute maximum along the channel dimension to normalize activations and handle long-tail outliers of video tokens, a method fundamentally derived from SmoothQuant[1] and ViDiT-Q[2]. VOLTS dynamically allocates training resources based on layer-wise variance, which is similar to SmoothQuant[1]. QAO employs a low-bit quantization branch alongside a full-precision low-rank branch, which is highly analogous to existing low-rank compensation methods like SVDQuant[3] and QuantVSR[4].
Overall, I suggest the authors refocus the narrative. Highlighting the specific algorithmic distinctions between LSGQuant and previous methods, or emphasizing the non-trivial efforts made to tailor these techniques for the one-step VSR task, would make for a much more compelling contribution.
2. **Ambiguous Formulation and Overselling of VOLTS**:
Ambiguous Formulation of VOLTS: The mathematical formulation of VOLTS lacks the precision required for reproducibility. Specifically, in Equations (5) and (6) , $\mu^l(X)$ averages the activation tensor only over the channel dimension $C$, leaving the batch $B$ and token $N$ dimensions intact. It remains entirely unclear how this multidimensional tensor is collapsed into the scalar sensitivity statistic $\sigma_l^2$ needed for Step 3's thresholding.
Overselling of VOLTS: Table 3(c) lacks decisive evidence to support the proposed three-tier classification. The method wins on PSNR (23.80 vs. 23.75) and DISTS by razor-thin margins, merely ties the best CLIP-IQA score (0.4966), and is outperformed on DOVER by the simplified baseline (0.4333 vs. 0.4571). Asserting better performance is an oversell; the results indicate the module is only marginally helpful rather than definitively validated
3. **Inconclusive Ablation of DRAQ**:
Table 3(a) fails to justify DRAQ's specific design. While the collapse of Calibrated Scaling (PSNR 20.04) shows that static approaches are brittle, it does not validate Equation (4). Furthermore, the razor-thin performance gap between No scaling (PSNR 23.72) and DRAQ (PSNR 23.80) merely suggests that online scaling is slightly helpful, falling short of proving the proposed adaptive mechanism is strictly necessary.
4. **Mathematical Inconsistencies and Presentation Issues**:
Equation (3) describes the quantized branch as utilizing the original rotated weight, denoted as $Q_A(XH)Q_W(HW)$. However, the immediately following text defines a residual weight, $W_R = W - L_1L_2$, and Algorithm 1 on Page 6 explicitly optimizes this residual rather than the rotated weight. This contradiction undermines the reproducibility of the method.
The presence of spelling mistakes and terminology slips, such as "Trining" and "Alternacing," while minor in isolation, collectively detract from the overall polish and professionalism of the submission.
---
[1] Xiao G, Lin J, Seznec M, et al. Smoothquant: Accurate and Efficient Post-training Quantization for Large Language Models[C]. ICML 2023.

[2] Zhao T, Fang T, Huang H, et al. VIDIT-Q: Efficient and Accurate Quantization of Diffusion Transformers for Image and Video Generation[C]. ICLR 2025.

[3] Li M, Lin Y, Zhang Z, et al. SVDQuant: Absorbing Outliers by Low-Rank Components for 4-Bit Diffusion Models [C]. ICLR 2025.

[4] Chai B, Chen Z, Zhu L, et al. QuantVSR: Low-Bit Post-Training Quantization for Real-World Video Super-Resolution[C]. AAAI 2026.

---

> ### Author Rebuttal · Authors · 2026-03-31
>
> # Response to Reviewer sphH
> `Q1-1` The three primary modules (DRAQ, VOLTS and QAO) are essentially a careful, task-specific synthesis of existing techniques.
>
> `A1-1` Thank you for the comment. We clarify the **novelty** of each component as follows.
>
> 1. **DRAQ**: DRAQ introduces an additional token‑wise scaling operation to mitigate outliers from both token and channel perspectives. Ablation results in Tab.4(a) show that it outperforms other quantization baselines (details discussed in `A1-3`).
>
> 2. **VOLTS**: While SmoothQuant migrates activation outliers to weights, VOLTS evaluates the sensitivity of each linear module to input variations and allocates training resources accordingly. Ablation on the UDM10 dataset (table below) shows that VOLTS improves performance. Further sensitivity-based grouping results in Tab.4(c) are discussed in `A1-2`.
>
> |Method|VOLTS|PSNR↑|DISTS↓|CLIP-IQA↑|DOVER↑|
> |-|-|-|-|-|-|
> |LSGQuant|| 21.76|0.3854|0.3038|0.2006|
> |LSGQuant|√(ours)|**23.80**|**0.2348**| **0.4966**|**0.4333**|
>
> 3. **QAO**: QAO is not a quantization architecture but a training‑strategy innovation. Moreover, the per‑module optimization strength from VOLTS is reflected in the number of iterations applied. Ablation in Tab.4(b) confirms its contribution.
>
> `Q1-2` The mathematical formulation of VOLTS lacks the precision required for reproducibility. What is the precise scalar statistic used to compute $\sigma_l^2$ in Eq.5/6? Tab.3(c) results indicate the module is only marginally helpful rather than definitively validated.
>
> `A1-2` Thank you for the comment.
>
> - In practice, we reshape the $B$ and $N$ into a single dimension, resulting in a $(\dots, C)$ tensor for each linear module. Then we compute variances along channel dimension and average them to get scalar scores.
>
> - We would like to clarify that the ablation study compares sensitivity‑based tiered optimization against the **Uniform Fully‑optimized baseline**. The `Frozen` and `Lightly Adapted` are additional configurations introduced after sensitivity‑based classification. The two‑tier and three‑tier variants show comparable or slightly improved performance, confirming the classification is reasonable. We have revised the table layout to present these results more clearly.
>
> `Q1-3` Ablation with DRAQ fails to justify its specific design.
>
> `A1-3` Thank you for the comment. We supplement the results in Tab.3(a) with additional perceptual metrics. This indicates that the proposed adaptive mechanism in Eq.(4) primarily enhances perceptual quality, which is not fully captured by PSNR.
>
> |Method|No Scaling|Calibrated Scaling|DRAQ(Ours)|
> |-|-|-|-|
> |LPIPS↓|0.3221|0.6462|**0.3169**|
> |MUSIQ↑|58.35|23.43|**59.14**|
> |MANIQA↑|0.2967|0.1760| **0.3139**|
> |$E^*_{warp}$↓|1.1456|7.8362| **1.0726**|
>
> `Q1-4` Eq.(3) has a contradiction with its context. Should the quantized branch use the residual weight $W_R$ rather than $W$? The presence of spelling mistakes and terminology slips collectively detracts from the overall polish and professionalism of the submission.
>
> `A1-4` Thank you for carefully pointing out the typo and notation issues. We acknowledge that this is a typo in Eq. (3), and the paper required closer attention to detail. We have thoroughly revised the camera‑ready version to correct all typos and improve notations.
>
> `Q1-5` How many layers fall into each of the three buckets for the reported experiments, and how sensitive are the results to $\delta_1$ and $\delta_2$?
>
> `A1-5` Thank you for the comment. For the 300 linear modules in the DiT blocks of WAN2.1, sensitivity scores naturally form three clusters:
> - **Low**: 90 layers, including all `k`, `v`, and `o` of cross-attention modules. $\sigma_l^2 \in [0, 3.03 \times 10^{-5}]$.
> - **Medium**: 193 layers, covering most modules in DiT. $\sigma_l^2 \in [1.88 \times 10^{-3}, 0.069]$.
> - **High**: 17 layers, mainly self‑attention modules near the beginning and end of DiT blocks. $\sigma_l^2 \in [0.089, 0.369]$.
>
> Because of clear numerical gaps, $\delta_1$ and $\delta_2$ can be chosen anywhere within the gaps without affecting the classification, making the method robust to the exact threshold values.
>
> `Q1-6` In Tab.1, were all baselines reimplemented or adapted under the same calibration dataset, optimization budget, and architectural compensation capacity?
>
> `A1-6` Thank you for the question. All baseline methods are reimplemented under fair conditions as far as possible. We provide detailed fairness protocols in the main paper and supplementary material.
> - We uniformly apply `HQ‑VSR` dataset for calibration.
> - ViDiT‑Q and SVDQuant are reimplemented from their official code. ViDiT‑Q’s codebase provides MinMax, SmoothQuant, and QuaRot.
> - All methods use asymmetric static weight quantization and symmetric dynamic activation quantization, following ViDiT-Q's setting.
> - For SmoothQuant and ViDiT‑Q, the pre‑scaling factor $\alpha$ is set to 0.5.
> - In SVDQuant, high‑precision branch rank is set to $r = 32$, consistent with LSGQuant.

---

> > ### Author Rebuttal · Reviewer_sphH · 2026-04-02
> >
> > Thank you for your comprehensive rebuttal.
> >
> > The additional ablation studies for DRAQ and VOLTS validated the effectiveness of your specific design, and your clear explanations of the technical details effectively resolved my previous confusion. I strongly encourage you to incorporate these supplementary results into your final revision and correct the typos (such as Equation 3) as promised. Additionally, it would be beneficial to the community if you could open-source your code for reproducibility.
> >
> > However, the authors' explanation regarding the novelty is not fully convincing, and I maintain that the core novelty might be considered modest. While the framework itself is effective for the VSR domain, I will keep my initial score for now and am open to further discussion with the other reviewers and the AC.

---

> > > ### Author Response · Authors · 2026-04-04
> > >
> > > # Further Discussion with Reviewer sphH
> > >
> > > Dear Reviewer sphH,
> > >
> > > Thank you for taking the time to review our response and for your thoughtful and constructive comments. We are glad that our rebuttal has addressed most of your concerns.
> > >
> > > - We sincerely appreciate your valuable suggestions and will refine our paper based on your feedback. We will also release our code soon to support full reproducibility.
> > >
> > > - Despite practical engineering efforts, our work also introduces **theoretical innovations**, whose effectiveness is validated by the ablation experiments in Tab.3:
> > >   1. `DRAQ` calculates both the channel's absolute maximum value and the token's relative absolute value. The serialization operation is friendly for quantization, allowing us to **suppress outliers** from both the token and channel perspectives.
> > >   2. `VOLTS` uses input variations as the metric for module sensitivity estimation. This variance-guided unified adaptation prioritizes perceptually critical components of the network. It determines training budgets with **simpler quantization settings and lower computational costs**.
> > >   3. `QAO` derives a training strategy that jointly optimizes a quantized branch and a full‑precision low‑rank branch, with early stopping and dynamically freezing insensitive layers. The strategy exhibits **a more principled joint optimization scheme** that deals with the coupling of the low-rank, high-precision branch and the quantized branch.
> > >
> > > We hope this response addresses your concern. Thank you again for your thoughtful feedback and kind support.
> > >
> > > Best regards,
> > >
> > > Authors

---

### Decision · Program_Chairs · 2026-04-30

**Decision:**

Accept (regular)

**Comment:**

This paper proposes LSGQuant, an effective quantization framework for one-step diffusion video super-resolution models. It combines dynamic range adaptation (DRAQ), variance-guided layer optimization (VOLTS), and joint quantization-aware training (QAO) to achieve over 70% parameter reduction and 3x inference speedups with near full-precision quality. Through a strong rebuttal featuring new comprehensive ablations, the authors successfully resolved reviewers' initial concerns regarding baseline fairness, mathematical clarity, and incremental novelty. Reviewers all agree that despite some concern on theoretical contributions, the framework's state-of-the-art empirical performance and clear practical deployment value well justify its acceptance.